# A Precompute-Then-Adapt Approach for Efficient Graph Condensation

## Abstract

Graph Neural Networks (GNNs) have shown great success in leveraging complex relationships in data but face significant computational challenges when dealing with large-scale graphs. To tackle this issue, graph condensation methods aim to compress large graphs into smaller, synthetic ones that can be efficiently used for GNN training. Recent approaches, particularly those based on trajectory matching, have achieved state-of-the-art (SOTA) performance in graph condensation tasks. Trajectory-based techniques match the training behavior on a condensed graph closely with that on the original graph, typically by guiding the trajectory of model parameters during training. However, these methods require repetitive re-training of GNNs during the condensation process, making them impractical for large graphs due to their high computational cost, *e.g.*, taking up to 22 days to condense million-node graphs. In this paper, we propose a novel Precompute-then-Adapt graph condensation framework that overcomes this limitation by separating the condensation process into a one-time precomputation stage and a one-time adaptation learning stage. Remarkably, even with only the precomputation stage, which typically takes seconds, our method surpasses or matches SOTA results on 3 out of 7 benchmark datasets. Extensive experiments demonstrate that our approach achieves better or comparable accuracy while being $96\times$ to $2,455\times$ faster in condensation time compared to SOTA methods, significantly enhancing the practicality of GNNs for large-scale graph applications. Our code and data are available at `https://anonymous.4open.science/r/GCPA-F6F9/`.

## 1 Introduction

Graph learning through Graph Neural Networks (GNNs) (Kipf & Welling, 2016; Hamilton et al., 2017) has significantly advanced graph data analysis, providing insights into complex structures in social networks (Fan et al., 2019; Zhang et al., 2022), molecular structures (Guo et al., 2021; Gasteiger et al., 2021), and beyond (Dong et al., 2023; Li & Zhu, 2021; Liu et al., 2020).

**Graph Condensation.** Large-scale graphs in real-world applications, often with millions of nodes and edges, pose significant computational challenges for training GNNs (Huang et al., 2021; Gao et al., 2024). Graph Condensation (GC) (Jin et al., 2021) generates a condensed graph (synthetic graph) from a large original graph, enabling models trained on the condensed graph to be directly applied to the original graph, achieving comparable performance on both graphs. GC enhances training efficiency by reducing computational costs associated with large-scale graphs. Recent studies demonstrate that GC facilitates efficient GNN training with minor performance loss (Gao et al., 2024; Jin et al., 2022; Yang et al., 2024; Liu et al., 2022; Zheng et al., 2024; Zhang et al., 2024).

**Structure-Free (SF) Condensation.** Structure-free methods have recently achieved state-of-the-art (SOTA) performance in node classification tasks (Zheng et al., 2024; Zhang et al., 2024). These methods condense the original graph into a new graph with only node features and labels, but with no edges. As illustrated in Figure 1a, the condensed graphs are structure-free, with nodes only connected to themselves. While it may be surprising that structure-free condensation can provide an effective summary of the original graph, these methods obtain SOTA performance and also simplify the optimization objectives compared to alternatives. Hence, we focus on structure-free condensation in our work.

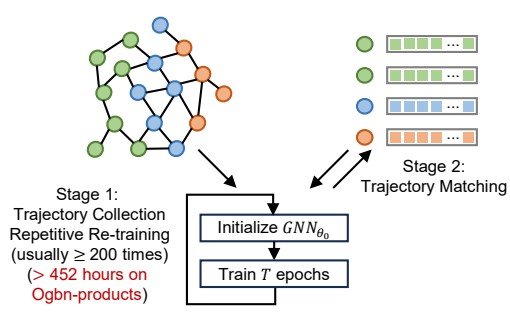 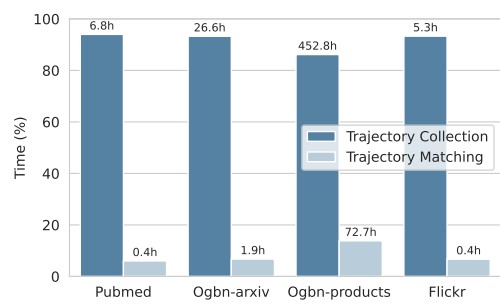

(a) Trajectory-based condensation

(b) Trajectory collection vs. trajectory matching time

Figure 1: Efficiency issue of trajectory-based methods. (a) Trajectory-based methods require repetitive GNN re-training during the trajectory collection stage, which can be highly time-consuming. (b) The trajectory collection stage takes the majority of running time in trajectory-based methods.

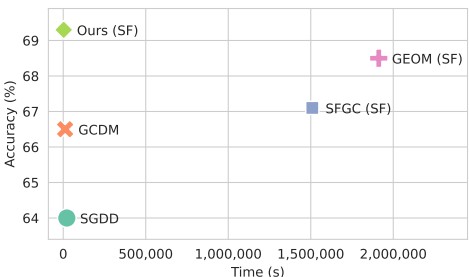

Figure 2: Performance vs. total condensation time on Ogbn-products dataset using GCN backbone. Our Precompute-then-Adapt framework employs a one-time precomputation and one-time adaptation learning stage, bypassing the time-intensive trajectory collection stage and achieving $96\times$ to $2,455\times$ condensation speed. Besides, our framework achieves superior performance compared to SOTA trajectory-based methods like GEOM and SFGC (SF: structure-free condensation methods).

**Trajectory-based Methods.** Trajectory matching has emerged as a key technique in recent advancements of graph condensation, as presented in methods SFGC (Zheng et al., 2024) and GEOM (Zhang et al., 2024). This approach assumes that the training trajectories, *i.e.*, the sequence of model parameters obtained by model training steps, should closely match for both the original and condensed graphs. The process starts by collecting training trajectories on the original graph. Then, the collected trajectories are used to set up the model on the condensed graph, aiming to keep the subsequent training steps consistent across both versions of the graph. This matching process ensures that the training is effective even on the condensed graph.

**Efficiency Issue of Trajectory-based Methods.** Trajectory-based methods require a substantial number of trajectories to achieve advanced performance, where each collected trajectory requires a complete training process that restarts from random parameter initialization. As depicted in Figure 1a, trajectory-based condensation requires repeating the model training process multiple times (e.g., 200 times), each time re-initializing the parameters and training the model for multiple epochs, taking a remarkably long time (e.g., 452 hours) on million-node graphs. As presented in Figure 1b, this stage takes up the majority of the total condensation time. This repetitive model re-training is a **key limitation** as it is highly time-consuming, resulting in extended running time, as shown in Figure 2. This inefficiency poses a significant barrier to applying these methods in practice, which can lead to missed opportunities in critical applications like social network analysis and fraud detection.

**Our Precompute-then-Adapt Method.** To address the inefficiencies of trajectory-based methods, we propose a novel Graph Condensation framework via a Precompute-then-Adapt approach (GCPA). Our method employs a **one-time precomputation stage** and **one-time adaptation learning stage**, eliminating the need for repetitive re-training with different random initializations. The precomputation stage involves extracting structural and semantic information from the original graph, achieving competitive performance within a short time. The adaptation stage further refines the precomputed features (representations) to improve performance with minor additional costs. As a result, we achieve competitive accuracy (-0.1% to +2.1%) on node classification tasks with substantially faster training time ($96\times$ to $2,455\times$) compared to SOTA trajectory-based methods.

Our two-stage precompute-then-adapt framework differs fundamentally from existing methods. With the help of precomputation of node features and adaptation of synthetic features, we achieve a level of computational efficiency that was previously unattainable. This framework significantly reduces training time while maintaining competitive performance, thereby opening opportunities for deploying graph condensation on resource-constrained devices.

We summarize the key contributions of our work as follows:

- We propose a new framework, GCPA, for graph condensation. It is **efficient**, consisting only of a one-time precomputation stage and a one-time adaptation learning stage. Compared to SOTA methods, our framework avoids costly repetitive re-training of models, achieving significant efficiency improvements.

- Our framework is also **effective**. With just the one-time precomputation stage, which extracts structural and semantic information from the original graph, our method can already surpass or match the performance of best baselines on 3 out of 7 benchmark datasets. With the one-time adaptation learning stage, we can further enhance performance via class-wise feature alignment, achieving SOTA results across all datasets.

- Through extensive experiments on benchmark datasets, we demonstrate that our method achieves better or comparable accuracy with up to $2{,}455\times$ faster training time than existing methods, making it more suitable for practical applications.

## 2 METHODOLOGY

In this section, we introduce the components and the implementation details of our framework.

### 2.1 PRELIMINARIES

Let $\mathcal{G} = \{\mathbf{X}, \mathbf{A}, \mathbf{Y}\}$ denotes a graph, where $\mathbf{X} \in \mathbb{R}^{N \times d}$ denotes the node features matrix with $N$ nodes with $d$-dimensional features, $\mathbf{A} \in \{0,1\}^{N \times N}$ represents the adjacency matrix encoding the graph structure, $\mathbf{Y} \in \mathbb{R}^{N \times C}$ denotes ground truth one-hot node labels on $C$ classes, while $\mathbf{y} \in \mathbb{R}^{N}$ is the label in vector form. Graph condensation aims at generating a synthetic graph corresponding to an existing graph such that a model trained on the synthetic graph is effective when applied to the original graph. Given an original graph $\mathcal{T} = \{\mathbf{X}, \mathbf{A}, \mathbf{Y}\}$ with $N$ nodes, the objective is to generate a smaller synthetic graph $\mathcal{S} = \{\mathbf{X}', \mathbf{A}', \mathbf{Y}'\}$ with $N'$ nodes such that a GNN trained on $\mathcal{S}$ achieves similar performance on $\mathcal{T}$ as another GNN trained directly on $\mathcal{T}$ for specific tasks. In particular, structure-free graph condensation emerges as a storage-efficient graph condensation approach where the adjacency matrix is set to an identity matrix, $\mathbf{A}' = \mathbf{I}$, so the synthetic graph does not contain structural information.

Node classification is a prevalent task simplified by graph condensation. This task involves assigning labels to nodes based on their features and structures. Formally, given a graph $\mathcal{G} = \{\mathbf{X}, \mathbf{A}\}$, and a subset of nodes $N_L \subseteq N$ with known labels $\mathbf{Y}_L \in \mathbb{R}^{N_L \times C}$, the transductive semi-supervised node classification task involves predicting labels $\mathbf{Y}_U \in \mathbb{R}^{N_U \times C}$ for an unlabeled subset of nodes $N_U \subseteq N$. The corresponding optimization goal can be formulated as a bi-level problem,

$$\min_{\mathcal{S}} \mathcal{L}(\text{GNN}_{\boldsymbol{\theta}_{\mathcal{S}}}(\mathbf{X}, \mathbf{A}), \mathbf{Y})$$
$$\text{s.t.} \quad \boldsymbol{\theta}_{\mathcal{S}} = \arg\min_{\boldsymbol{\theta}} \mathcal{L}(\text{GNN}_{\boldsymbol{\theta}}(\mathbf{X}', \mathbf{A}'), \mathbf{Y}'), \tag{1}$$

where $\boldsymbol{\theta}$ denotes the learnable parameters of a GNN model, $\boldsymbol{\theta}_{\mathcal{S}}$ represents the optimal GNN parameters learned on the synthetic graph, $\mathcal{L}$ is a loss function evaluating the node classification performance. Existing graph condensation approaches optimize this bi-level problem to learn an optimal synthetic graph $\mathcal{S}$ such that a trained GNN with parameters $\boldsymbol{\theta}_{\mathcal{S}}$ yields optimal performance on $\mathcal{T}$. However, the bi-level optimization problem is computationally intensive as it involves nested optimization loops. To mitigate this issue, we introduce our framework that directly optimizes synthetic node features for improved efficiency.

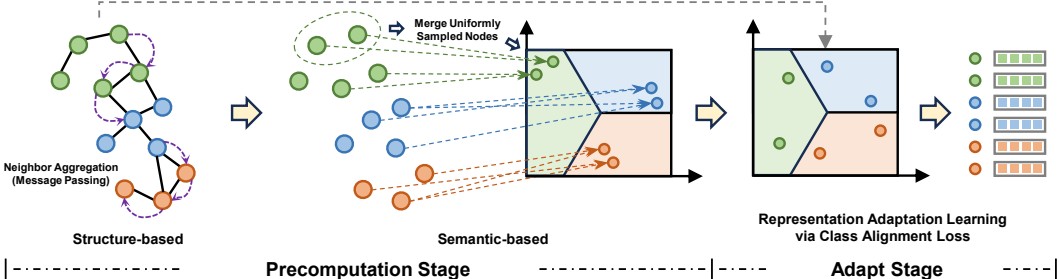

Figure 3: Overall pipeline of the proposed GCPA condensation framework.

## 2.2 OVERVIEW OF GCPA FRAMEWORK

The overall pipeline of the proposed precompute-then-adapt graph condensation framework is provided in Figure 3. We introduce two stages, *i.e.*, a precomputation stage and a representation adaptation learning stage to produce structure-free synthetic data. The precomputation stage involves structure-based neighbor aggregation and semantic-based merging on the original graph, which achieves competitive performance within a relatively short time. The representation adaptation learning stage further refines the precomputed features using a class-wise feature alignment objective to improve performance with minor additional costs.

## 2.3 STRUCTURE-BASED PRECOMPUTATION

In the context of graph-based learning models, neighbor information aggregation refers to the process by which node features are enriched with the structural information from neighboring nodes. This process allows a node's feature vector to incorporate not just its own information but also that of its surrounding neighborhood. This aggregation is critical for capturing relationships and dependencies in graph-structured data.

Drawing inspiration from the graph diffusion process (Gasteiger et al., 2019), we leverage neighbor structural information to pre-process the original node features. The goal of graph diffusion is to smooth node features based on the underlying graph's topology, facilitating the effective propagation of information across nodes. The structure-based precomputed features $\mathbf{H}$ with $K$-hop neighbor aggregation can be recursively computed as:

$$\mathbf{H}^{(k)} = (1 - \alpha)\hat{\mathbf{A}}\mathbf{H}^{(k-1)} + \alpha\mathbf{H}^{(0)}, \quad \text{for } k = 1, 2, \ldots, K,$$
$$\text{with} \quad \hat{\mathbf{A}} = \tilde{\mathbf{D}}^{-\frac{1}{2}}\tilde{\mathbf{A}}\tilde{\mathbf{D}}^{-\frac{1}{2}}, \quad \tilde{\mathbf{A}} = \mathbf{A} + \mathbf{I}_N, \tag{2}$$

where $\mathbf{H}^{(0)} = \mathbf{X}$ represents the node feature matrix, $K$ denotes the number of aggregation hops, $\mathbf{H} = \mathbf{H}^{(K)}$ is the output of the last layer, coefficient $\alpha$ controls the contribution of raw features to each hop. Having processed the structural information, we omit the edges in the follow-up semantic-based precomputation as shown in Figure 3, focusing on processing semantic information.

## 2.4 SEMANTIC-BASED PRECOMPUTATION

To condense a set of $N$ aggregated features into $N'$ synthetic node features, we perform semantic-based precomputation by merging uniformly sampled original nodes within each class. This approach ensures that each synthetic node represents the core semantic characteristics of its class in the synthetic dataset.

Specifically, for each synthetic node $v_i$ with class label $c \in \{1, 2, \ldots, C\}$, we uniformly sample a subset of original nodes in the same class. Then, we compute the semantic-based features by taking the mean of the aggregated features of the sampled nodes:

$$\hat{\mathbf{X}}'_i = \frac{1}{M} \sum_{j \in \mathcal{S}_i} \mathbf{H}_j, \quad \text{for } i = 1, 2, \ldots, N',$$
$$\text{s.t.} \quad \mathcal{S}_i \subseteq \mathcal{I}_{\mathbf{y}_i}, \quad |\mathcal{S}_i| = M, \quad \mathcal{I}_c = \{i \mid \mathbf{y}_i = c\}, \tag{3}$$

where $\mathcal{S}_i$ is the set of sampled original nodes for synthetic node $i$, $\mathcal{I}_c$ denotes the indices of original nodes belonging to class $c$, $M$ is the number of sampled nodes for each synthetic node.

This semantic-based precomputation process effectively condenses the semantic information of multiple nodes within the same class into a single synthetic node. Furthermore, by maintaining the class distribution in $\mathbf{Y}$ through proportional sampling, we fix the synthetic labels $\mathbf{Y}'$ to preserve the original class proportions. Consequently, we obtain the precomputed condensed dataset $\{\hat{\mathbf{X}}', \mathbf{Y}'\}$, where $\hat{\mathbf{X}}' \in \mathbb{R}^{N' \times d}$ and $\mathbf{Y}' \in \mathbb{R}^{N' \times C}$.

## 2.5 REPRESENTATION ADAPTATION LEARNING

Given the limited number of condensed nodes, it is crucial that these nodes ideally depict the overall representations of their respective classes (depicted by the background color in Figure 3). Although the precomputation stage focuses on capturing the structural and semantic information of the original graph, its non-learning process could lead to sub-optimal representations to depict the class-wise overall representations, as illustrated in Figure 3.

To address this limitation, we introduce a representation adaptation learning stage to further refine the precomputed representations. We notice that representation contrastive loss can be considered to enhance node embeddings for improved classification utility (Joshi et al., 2022). In the context of graph condensation, we propose to align the condensed features with the original precomputed features using a class-wise representation adaptation objective.

Specifically, we introduce an adaptation module $f_{\text{adapt}} : \mathbb{R}^{N' \times d} \to \mathbb{R}^{N' \times d}$, implemented as a Multi-Layer Perceptron (MLP), to adapt the synthetic features to better depict the overall representations:

$$
\begin{aligned}
\mathbf{Z}' &= \beta \hat{\mathbf{X}}' + (1 - \beta) f_{\text{adapt}}(\hat{\mathbf{X}}') \\
&= \beta \hat{\mathbf{X}}' + (1 - \beta) \text{MLP}(\hat{\mathbf{X}}'),
\end{aligned}
\tag{4}
$$

where $\beta$ is a hyperparameter controlling the contribution of precomputed representations, $\mathbf{Z}'$ represents the adapted synthetic representations. We adopt $\mathbf{X}' = \mathbf{Z}'$ after the learning process.

We further construct the contrastive samples by first sampling a sufficient number of nodes as anchors from the precomputed representations, from which the adaptation module learns to refine the synthetic representations. For each anchor node on the original graph, we sample a synthetic node belonging to the same class as a positive sample and a set of arbitrary synthetic nodes as negative samples. With the sampled contrastive pairs, we optimize the cross entropy loss to distinguish between the adapted positive and negative samples:

$$
\mathcal{L} = -\mathbb{E}_{\{i,j|\mathbf{y}_i = \mathbf{y}'_j\}} \left( \log \sigma\Big(\langle \mathbf{H}_i, \mathbf{Z}'_j \rangle\Big) + \frac{1}{S} \sum_{s=1}^{S} \mathbb{E}_{t \sim \text{Uniform}\{1,...,N'\}} \log \sigma\Big(-\langle \mathbf{H}_i, \mathbf{Z}'_t \rangle\Big) \right)
\tag{5}
$$

where $\langle \mathbf{H}_i, \mathbf{Z}'_j \rangle$ computes the inner product between $i$-th anchor node's representation $\mathbf{H}_i$ and $j$-th synthetic node's adapted representation $\mathbf{Z}'_j$, $S$ denotes the number of negative samples for an anchor node, $t$ is the index of a random negative sample on the synthetic dataset, $\sigma(x) = 1/(1 + \exp(-x))$ is the sigmoid function. The adaptation module $f_{\text{adapt}}$ refines the precomputed representations, achieving better alignment of the overall representations between the synthetic and original graphs, and improving the generalization of the condensed features.

## 3 EXPERIMENTS

In this section, we conduct experiments to validate the effectiveness of the proposed framework.

### 3.1 EXPERIMENTAL SETUP

**Datasets.** Following GCondenser (Liu et al., 2024), a comprehensive graph condensation benchmark, our experiments are conducted on seven benchmark datasets including three smaller networks: CiteSeer, Cora, and PubMed (Kipf & Welling, 2016), and four larger graphs: Ogbn-arxiv, Ogbn-products (Hu et al., 2020), Flickr (Zeng et al., 2019), and Reddit (Hamilton et al., 2017). We use the public data splits for fair comparisons. The dataset statistics and settings are detailed in Table 1. For CiteSeer, Cora, and PubMed datasets, row feature normalization is applied to prepare the data. For Ogbn-arxiv, Flickr, and Reddit datasets, we apply feature standardization. The Ogbn-products dataset retains its feature processing as defined by OGB (Hu et al., 2020).

Table 1: Dataset statistics. (Trans.: transductive. Ind.: inductive. #Feat.: number of features. #Cls.: number of classes.)

| Setting | Dataset | #Train/Val/Test Nodes | #Nodes | #Edges | #Feat. | #Cls. |
|---|---|---|---|---|---|---|
| Trans. | Citeseer | 120/500/1,000 | 3,327 | 4,732 | 3,703 | 6 |
| | Cora | 140/500/1,000 | 2,708 | 5,429 | 1,433 | 7 |
| | PubMed | 60/500/1,000 | 19,717 | 88,648 | 500 | 3 |
| | Ogbn-arxiv | 90,941/29,799/48,603 | 169,343 | 1,166,243 | 128 | 40 |
| | Ogbn-products | 196,615/39,323/2,213,091 | 2,449,029 | 61,859,140 | 100 | 47 |
| Ind. | Flickr | 44,625/22,312/22,313 | 89,250 | 899,756 | 500 | 7 |
| | Reddit | 153,431/23,831/55,703 | 232,965 | 57,307,946 | 602 | 41 |

**Baselines.** We compare our proposed framework to the baselines in the following categories: (i) Coreset approach: K-Center (Sener & Savarese, 2017). (ii) Gradient matching approaches: GCond (Jin et al., 2021) and SGDD (Yang et al., 2024). (iii) Distribution matching approach: GCDM (Liu et al., 2022). (iv) Trajectory matching approaches: SFGC (Zheng et al., 2024) and GEOM (Zhang et al., 2024). We use the implementations provided by GCondenser (Liu et al., 2024) for fair comparisons between our method and the baselines.

**Backbone Models.** We use GCN (Kipf & Welling, 2016) and SGC (Wu et al., 2019) as backbone models during condensation and evaluation for fair comparisons. In the cross-architecture evaluation, we use more backbones including GAT (Veličković et al., 2018), ChebNet (Defferrard et al., 2016), GraphSAGE (Hamilton et al., 2017), and APPNP (Gasteiger et al., 2018).

**Evaluation.** Following GCondenser (Liu et al., 2024), we evaluate all methods using three different condensation ratios (r) for each dataset. Specifically, the condensation ratio r is defined as the fraction of condensed nodes rN to the total number of original nodes $N$, where $0 < r < 1$. In the transductive setting, N denotes the total node count in the entire large-scale graph, whereas in the inductive setting, N refers to the node count within the training sub-graph of the complete large-scale graph. The evaluation has two phases: (i) the condensation phase: synthesizes the condensed graph from the original graph, and (ii) the evaluation phase: the GNN is trained on the condensed graph, and the performance is evaluated on the original test nodes. We repeat the experiments five times and report the average node classification accuracy with standard deviation. The experiments are conducted on a single NVIDIA H100 GPU (80GB).

**Hyper-parameter Settings.** We tune the strcuture-based precomputation hops $K \in \{1, 2, 3, 4\}$, damping factor $\alpha \in \{0, 0.25, 0.5, 0.75\}$, residual coefficient $\beta \in \{0, 0.25, 0.5, 0.75\}$, semantic-based aggregation size $M \in \{1, 10, 50, 100\}$, number of negative samples $S \in \{1, 5, 10, 50\}$, number of adaptation layers $\{1, 2, 3\}$, hidden dimension of the adaptation module $\{128, 256, 512\}$. We tune all hyper-parameter on the validation set. To ensure fair comparisons, we follow GCondenser (Liu et al., 2024) to set the number of backbone model layers to 2, hidden dimension to 256, weight decay to 0.0005, dropout rate to 0.5, and learning rate to 0.01.

## 3.2 PERFORMANCE COMPARISON

We present the performance of different graph condensation approaches using the GCN backbone in Table 2. Additionally, the performance of these approaches with the SGC backbone is shown in Table 6, located in the Appendix. Based on these results, we make the following observations:

- The coreset approach, K-Center, which typically employs conventional machine learning techniques, fails to provide good condensation results on all datasets. This highlights the non-trivial nature of graph condensation tasks, which necessitate substantial effort.

- Two distinct categories of graph condensation methods, including gradient matching with GCond and SGDD, and distribution matching with GCDM, have both shown fair performance on different datasets. It is worth noting that neither category consistently outperforms the other across all datasets. This variation in performance suggests that multiple frameworks might be applicable for the task of graph condensation, without a universally superior approach.

- Recent advancements in trajectory matching, especially the SFGC and GEOM approaches, have demonstrated superior performance on most datasets, affirming the efficacy of trajectory-based

Table 2: Node classification performance comparison using GCN backbone (mean±std). The best and second-best results are marked in bold and underlined, respectively. Ours (Pre.) is our precomputation-only variant. The Whole column represents the performance obtained by training on the whole dataset.

| Dataset | Ratio | K-Cen. | GCond | SGDD | GCDM | SFGC | GEOM | Ours (Pre.) | Ours | Whole |
|---|---|---|---|---|---|---|---|---|---|---|
| Citeseer | 0.9% | $65.0_{\pm 0.0}$ | $46.3_{\pm 7.0}$ | $70.6_{\pm 1.5}$ | $71.2_{\pm 0.8}$ | $69.7_{\pm 0.3}$ | $69.6_{\pm 0.6}$ | $\underline{72.1}_{\pm 0.2}$ | $\mathbf{72.4}_{\pm 0.4}$ | |
| | 1.8% | $67.8_{\pm 0.0}$ | $54.2_{\pm 3.9}$ | $71.5_{\pm 0.7}$ | $71.9_{\pm 0.7}$ | $69.4_{\pm 0.0}$ | $67.5_{\pm 0.9}$ | $\underline{72.1}_{\pm 0.1}$ | $\mathbf{72.9}_{\pm 0.3}$ | $71.4_{\pm 0.5}$ |
| | 3.6% | $69.4_{\pm 0.0}$ | $70.7_{\pm 0.7}$ | $71.0_{\pm 0.7}$ | $72.3_{\pm 1.3}$ | $69.8_{\pm 0.5}$ | $72.1_{\pm 1.0}$ | $\underline{72.7}_{\pm 0.5}$ | $\mathbf{72.8}_{\pm 0.1}$ | |
| Cora | 1.3% | $66.5_{\pm 0.0}$ | $\underline{80.5}_{\pm 0.4}$ | $80.5_{\pm 0.4}$ | $78.9_{\pm 0.8}$ | $79.6_{\pm 0.2}$ | $80.3_{\pm 1.1}$ | $80.3_{\pm 0.5}$ | $\mathbf{81.4}_{\pm 0.6}$ | |
| | 2.6% | $71.6_{\pm 0.0}$ | $\underline{78.1}_{\pm 3.6}$ | $81.2_{\pm 0.6}$ | $79.4_{\pm 0.6}$ | $79.5_{\pm 0.1}$ | $81.5_{\pm 0.8}$ | $80.6_{\pm 0.5}$ | $\mathbf{81.9}_{\pm 1.0}$ | $81.7_{\pm 0.9}$ |
| | 5.2% | $76.6_{\pm 0.0}$ | $80.2_{\pm 1.7}$ | $79.9_{\pm 1.6}$ | $79.9_{\pm 0.2}$ | $80.1_{\pm 0.6}$ | $\underline{82.2}_{\pm 0.4}$ | $80.8_{\pm 0.3}$ | $\mathbf{82.4}_{\pm 0.7}$ | |
| PubMed | 0.08% | $72.1_{\pm 0.1}$ | $67.6_{\pm 0.4}$ | $76.7_{\pm 1.1}$ | $75.9_{\pm 0.6}$ | $78.4_{\pm 0.1}$ | $80.1_{\pm 0.3}$ | $79.5_{\pm 1.3}$ | $\mathbf{80.2}_{\pm 1.9}$ | |
| | 0.15% | $76.4_{\pm 0.0}$ | $74.6_{\pm 0.8}$ | $78.5_{\pm 0.4}$ | $77.4_{\pm 0.4}$ | $78.1_{\pm 0.4}$ | $\underline{79.7}_{\pm 0.3}$ | $79.7_{\pm 0.3}$ | $\mathbf{80.5}_{\pm 0.8}$ | $79.3_{\pm 0.3}$ |
| | 0.3% | $78.2_{\pm 0.0}$ | $77.2_{\pm 0.7}$ | $78.0_{\pm 1.1}$ | $77.6_{\pm 0.4}$ | $78.5_{\pm 0.5}$ | $\underline{79.5}_{\pm 0.4}$ | $79.3_{\pm 0.3}$ | $\mathbf{81.6}_{\pm 2.4}$ | |
| Arxiv | 0.05% | $54.5_{\pm 0.0}$ | $53.7_{\pm 1.6}$ | $55.9_{\pm 5.8}$ | $63.3_{\pm 0.3}$ | $\underline{66.1}_{\pm 0.4}$ | $65.5_{\pm 1.0}$ | $60.5_{\pm 0.9}$ | $\mathbf{67.2}_{\pm 0.3}$ | |
| | 0.25% | $60.3_{\pm 0.0}$ | $64.2_{\pm 0.2}$ | $63.2_{\pm 0.3}$ | $66.4_{\pm 0.1}$ | $\underline{67.2}_{\pm 0.4}$ | $65.8_{\pm 0.4}$ | $64.6_{\pm 0.4}$ | $\mathbf{67.7}_{\pm 0.2}$ | $71.1_{\pm 0.0}$ |
| | 0.5% | $62.1_{\pm 0.0}$ | $65.1_{\pm 0.4}$ | $66.8_{\pm 0.3}$ | $67.6_{\pm 0.0}$ | $\underline{67.8}_{\pm 0.2}$ | $66.2_{\pm 0.5}$ | $65.5_{\pm 0.3}$ | $\mathbf{68.1}_{\pm 0.3}$ | |
| Products | 0.025% | $55.4_{\pm 0.8}$ | $63.7_{\pm 0.3}$ | $64.0_{\pm 0.4}$ | $66.5_{\pm 0.1}$ | $67.1_{\pm 0.2}$ | $\underline{68.5}_{\pm 0.3}$ | $64.1_{\pm 0.9}$ | $\mathbf{69.3}_{\pm 0.2}$ | |
| | 0.05% | $57.6_{\pm 0.7}$ | $67.0_{\pm 0.2}$ | $65.9_{\pm 0.2}$ | $68.4_{\pm 0.4}$ | $67.9_{\pm 0.3}$ | $\underline{69.8}_{\pm 0.3}$ | $65.9_{\pm 0.9}$ | $\mathbf{69.9}_{\pm 0.7}$ | $73.1_{\pm 0.1}$ |
| | 0.1% | $59.1_{\pm 0.5}$ | $68.0_{\pm 0.2}$ | $66.1_{\pm 0.3}$ | $68.4_{\pm 0.3}$ | $70.1_{\pm 0.3}$ | $\underline{71.1}_{\pm 0.3}$ | $67.7_{\pm 0.3}$ | $\mathbf{71.3}_{\pm 0.7}$ | |
| Flickr | 0.1% | $40.7_{\pm 0.0}$ | $43.3_{\pm 0.3}$ | $42.4_{\pm 0.2}$ | $44.5_{\pm 0.4}$ | $\underline{46.9}_{\pm 0.3}$ | $44.6_{\pm 0.0}$ | $44.4_{\pm 0.4}$ | $\mathbf{47.0}_{\pm 0.2}$ | |
| | 0.5% | $41.4_{\pm 0.0}$ | $44.6_{\pm 0.4}$ | $44.9_{\pm 0.3}$ | $45.0_{\pm 0.2}$ | $\underline{47.0}_{\pm 0.1}$ | $45.2_{\pm 0.9}$ | $45.4_{\pm 0.1}$ | $\mathbf{47.1}_{\pm 0.1}$ | $46.8_{\pm 0.2}$ |
| | 1% | $41.4_{\pm 0.0}$ | $44.4_{\pm 0.1}$ | $45.2_{\pm 0.2}$ | $44.6_{\pm 0.3}$ | $\mathbf{47.2}_{\pm 0.1}$ | $45.5_{\pm 0.1}$ | $45.4_{\pm 0.1}$ | $\underline{47.1}_{\pm 0.1}$ | |
| Reddit | 0.05% | $58.6_{\pm 0.1}$ | $56.8_{\pm 2.1}$ | $72.9_{\pm 4.9}$ | $88.9_{\pm 1.2}$ | $89.2_{\pm 0.5}$ | $90.0_{\pm 0.5}$ | $\underline{90.5}_{\pm 0.3}$ | $\mathbf{90.5}_{\pm 0.3}$ | |
| | 0.1% | $81.7_{\pm 0.0}$ | $87.4_{\pm 0.4}$ | $89.6_{\pm 2.5}$ | $\underline{91.8}_{\pm 0.3}$ | $90.9_{\pm 0.3}$ | $89.4_{\pm 0.5}$ | $91.3_{\pm 0.2}$ | $\mathbf{92.4}_{\pm 0.1}$ | $94.2_{\pm 0.0}$ |
| | 0.2% | $86.9_{\pm 0.0}$ | $91.4_{\pm 0.4}$ | $91.2_{\pm 0.3}$ | $92.2_{\pm 0.1}$ | $\underline{92.4}_{\pm 0.1}$ | $91.2_{\pm 0.1}$ | $91.4_{\pm 0.1}$ | $\mathbf{92.9}_{\pm 0.2}$ | |

Table 3: Efficiency comparison using GCN backbone (total condensation time in seconds).

| Dataset | K-Center | GCond | SGDD | GCDM | SFGC | GEOM | Ours (Pre.) | Ours |
|---|---|---|---|---|---|---|---|---|
| Citeseer (r=1.8%) | 7 | 71 | 70 | 57 | 2,165 | 10,890 | 6 | 45 |
| Cora (r=2.6%) | 5 | 70 | 70 | 54 | 2,578 | 10,144 | 4 | 44 |
| PubMed (r=0.15%) | 5 | 59 | 223 | 48 | 8,060 | 26,432 | 5 | 39 |
| Arxiv (r=0.25%) | 18 | 389 | 759 | 555 | 86,553 | 104,905 | 20 | 247 |
| Products (r=0.05%) | 91 | 13,554 | 21,821 | 11,485 | 1,509,397 | 1,912,105 | 104 | 2,985 |
| Flickr (r=0.5%) | 16 | 187 | 1,178 | 165 | 96,350 | 21,061 | 23 | 219 |
| Reddit (r=0.1%) | 51 | 2,665 | 12,126 | 1563 | 379,974 | 128,642 | 55 | 505 |

methods. Notably, both SFGC and GEOM employ structure-free condensation, indicating that for node classification tasks, providing edges in condensed graphs may not always be necessary.

- Our proposed framework achieves state-of-the-art performance in 40 out of 42 condensation settings using GCN and SGC backbones, underscoring the effectiveness of our precompute-then-adapt approach, which is a novel graph condensation framework different from existing methods.

## 3.3 Efficiency Comparison

We present a comprehensive efficiency comparison of different methods using the GCN backbone in Table 3 and the SGC backbone in Table 7 (located in the Appendix). Additionally, Figure 4 illustrates a joint analysis of both accuracy and efficiency. Based on the results, we make the following observations on the efficiencies of different approaches:

- As depicted in Figure 4, trajectory-based methods including SFGC and GEOM exhibit leading performance but suffer from poor efficiency. The primary efficiency bottleneck lies in the need for repetitive re-training, which, while effective, leads to severe efficiency issues.

- As shown in Figure 4, our framework achieves state-of-the-art performance across all presented datasets. Notably, the framework is significantly more efficient than trajectory-based methods, achieving speedups ranging from $96\times$ to $2,455\times$ compared to the trajectory-based approaches.

- Our method is not only more efficient than the time-intensive trajectory-based methods but also faster than the majority of other baseline methods on most datasets. These results underscore the superior condensation efficiency of our precompute-then-adapt framework.

- A variant of our method containing only the precomputation stage (Pre.), typically taking under 60 seconds to complete, matches or surpasses the performance of 3 out of 7 datasets, as detailed in Table 2. The presented results illustrate the capability of the precomputation stage to achieve competitive results in a fraction of the time compared to learning-based baselines.

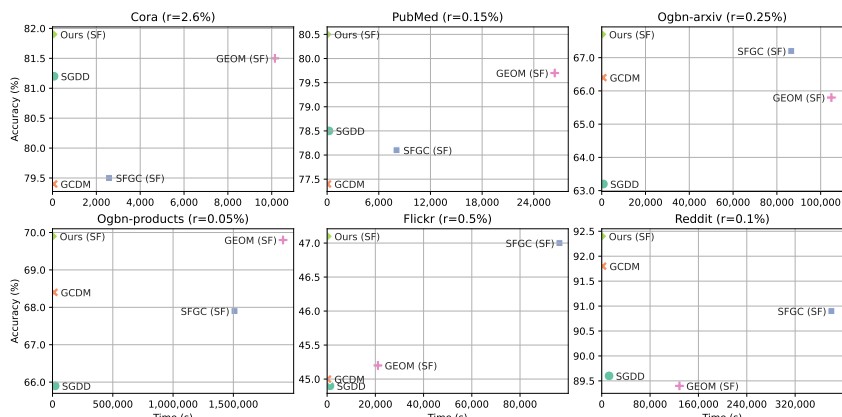

Figure 4: Evaluation accuracy *vs.* total condensation time using GCN backbone.

Table 4: Ablation study on components in the precomputation stage including strcuture-based and semantic-based aggregation phases. The best results are marked in bold.

| Dataset | Full | w/o Structural | w/o Semantic | w/o Both |
|---|---|---|---|---|
| Citeseer (r=1.80%) | **72.9** | 69.3 | 67.8 | 62.1 |
| Cora (r=2.60%) | **81.9** | 74.4 | 79.1 | 72.2 |
| PubMed (r=0.15%) | **80.5** | 77.9 | 78.8 | 76.1 |
| Arxiv (r=0.25%) | **67.7** | 64.2 | 67.3 | 63.9 |
| Products (r=0.05%) | **69.9** | 64.6 | 65.7 | 62.2 |
| Flickr (r=0.5%) | **47.1** | 46.9 | 47.0 | 46.9 |
| Reddit (r=0.10%) | **92.4** | **92.4** | 92.2 | 92.2 |

These observations demonstrate that our method not only achieves competitive performance but does so with markedly higher efficiency, addressing one of the key challenges in scalable graph learning applications.

### 3.4 ABLATION

Table 4 evaluates the impact of structure-based and semantic-based phases of the precomputation stage. The results show that both the structural and semantic components contribute to the performance of the framework, particularly on transductive datasets, which reflects the importance of precomputation on transductive datasets where the complete graph structure is available. We also observe that the removal of structural components typically results in a larger performance drop compared to the removal of semantic components. This indicates the critical role of structure-based aggregation in capturing representative features in the original graph. In conclusion, the structural and semantic components are both pivotal to achieving optimal performance in our framework, but their impact varies with the nature of the datasets.

### 3.5 CROSS-ARCHITECTURE TRANSFERABILITY

Table 5 presents the cross-architecture transferability results of condensed graphs across different models. Our method consistently outperforms or matches the top performance across all datasets, underscoring the robustness and generalization of our framework. The ability to transfer across different architectures may be attrbuted to the similar filtering behaviors of popular GNNs, as reported in existing literature (Jin et al., 2021; Zheng et al., 2024). In particular, our framework demonstrates outstanding transferability, which may be attributed to our direct alignment between original and synthetic features, without relying on specific GNN models for performance matching.

### 3.6 VISUALIZATION

Figure 5 displays visualization results between SFGC condensed features and ours. Our condensed graphs exhibit clear clustering patterns on all presented datasets with minimal inter-class mixing, in

Table 5: Cross-architecture transferability of condensed graphs using GCN backbone. The best and second-best results are marked in bold and underlined, respectively.

| Dataset | Method | MLP | SGC | GCN | GAT | ChebNet | SAGE | APPNP | Avg. | Std. |
|---|---|---|---|---|---|---|---|---|---|---|
| Citeseer ($r = 0.90\%$) | GCond | 41.8 | 34.8 | 46.3 | 39.2 | 57.4 | 61.2 | 47.0 | 46.8 | 8.8 |
| | GCDM | 62.3 | 69.6 | 72.7 | 58.3 | 60.2 | 67.1 | 71.4 | 65.9 | 5.3 |
| | SFGC | 64.4 | 64.9 | 70.4 | 70.0 | 69.1 | 69.5 | 70.8 | 68.4 | 2.5 |
| | Ours | 66.5 | 70.9 | 73.4 | 73.4 | 72.8 | 72.6 | 72.1 | **71.7** | 2.3 |
| Cora ($r = 1.30\%$) | GCond | 67.7 | 72.6 | 79.5 | 80.7 | 60.0 | 78.6 | 79.0 | 74.0 | 7.2 |
| | GCDM | 65.3 | 78.5 | 80.2 | 80.1 | 58.4 | 77.5 | 79.3 | 74.2 | 8.1 |
| | SFGC | 68.2 | 76.2 | 80.4 | 79.8 | 62.1 | 77.6 | 81.6 | 75.1 | 6.7 |
| | Ours | 70.5 | 79.9 | 81.3 | 79.1 | 82.1 | 78.9 | 76.2 | **78.3** | 3.6 |
| Pubmed ($r = 0.08\%$) | GCond | 75.1 | 55.6 | 75.0 | 77.0 | 74.3 | 77.2 | 78.0 | 73.2 | 7.3 |
| | GCDM | 73.8 | 72.9 | 75.0 | 73.7 | 70.5 | 75.3 | 76.9 | 74.0 | 1.9 |
| | SFGC | 73.6 | 76.8 | 78.5 | 76.6 | 77.2 | 76.7 | 78.9 | **76.9** | 1.6 |
| | Ours | 74.2 | 76.6 | 76.1 | 76.3 | 77.3 | 77.5 | 76.7 | 76.4 | 1.0 |
| Arxiv ($r = 0.05\%$) | GCond | 39.2 | 58.0 | 57.0 | 47.7 | 36.4 | 33.5 | 54.3 | 46.6 | 9.5 |
| | GCDM | 41.6 | 59.8 | 60.7 | 46.5 | 52.6 | 55.3 | 60.3 | 53.8 | 6.9 |
| | SFGC | 45.3 | 62.2 | 63.3 | 60.5 | 50.7 | 55.4 | 62.4 | 57.1 | 6.4 |
| | Ours | 46.7 | 61.6 | 65.0 | 64.4 | 63.3 | 58.4 | 53.9 | **59.0** | 6.2 |
| Products ($r = 0.025\%$) | GCond | 36.4 | 45.7 | 60.7 | 48.4 | 45.2 | 49.8 | 60.3 | 49.5 | 8.0 |
| | GCDM | 45.7 | 60.0 | 66.6 | 67.9 | 61.2 | 63.6 | 66.2 | 61.6 | 7.0 |
| | SFGC | 46.7 | 55.1 | 66.7 | 69.4 | 61.4 | 63.4 | 64.8 | 61.1 | 7.2 |
| | Ours | 46.3 | 65.9 | 65.9 | 67.6 | 67.8 | 62.2 | 62.1 | **62.5** | 7.0 |
| Flickr ($r = 0.1\%$) | GCond | 40.8 | 36.5 | 44.9 | 40.8 | 43.0 | 43.2 | 44.9 | 42.0 | 2.7 |
| | GCDM | 41.7 | 27.3 | 40.7 | 37.7 | 41.5 | 43.0 | 43.8 | 39.4 | 5.3 |
| | SFGC | 44.9 | 38.7 | 46.2 | 45.3 | 43.6 | 44.9 | 46.2 | 44.3 | 2.4 |
| | Ours | 44.1 | 45.1 | 45.3 | 45.1 | 42.4 | 43.6 | 45.4 | **44.4** | 1.0 |
| Reddit ($r = 0.05\%$) | GCond | 38.7 | 82.2 | 79.9 | 31.2 | 38.7 | 41.5 | 69.8 | 54.6 | 20.2 |
| | GCDM | 43.1 | 87.1 | 88.1 | 37.5 | 55.6 | 66.2 | 68.9 | 63.8 | 18.3 |
| | SFGC | 47.5 | 82.8 | 87.0 | 84.4 | 53.6 | 71.9 | 67.5 | 70.7 | 14.4 |
| | Ours | 39.3 | 91.1 | 90.9 | 90.5 | 61.8 | 79.1 | 66.4 | **74.1** | 18.1 |

contrast to the SFGC graphs which show less distinct class separation. The comparison is more evident on larger datasets such as Ogbn-arxiv and Flickr, where SFGC fails to produce clear clustering patterns. To quantify these clustering patterns, we follow previous work (Zhang et al., 2024) to utilize clustering metrics including the Silhouette Coefficient (Rousseeuw, 1987), the Davies-Bouldin Index (Davies & Bouldin, 1979), and the Calinski-Harabasz Index (Caliński & Harabasz, 1974), all of which indicate that our condensed graphs demonstrate better clustering patterns. The visualization results show that our framework effectively optimizes the condensed features, forming robust representations to preserve the original graph's classification capabilities.

# 4 RELATED WORK

**Graph Condensation.** Graph condensation (Jin et al., 2021; 2022; Yang et al., 2023; Liu et al., 2022; Xu et al., 2024), derived from dataset distillation, is a technique aimed at producing a much smaller version of a graph while retaining as much information as possible from the original. Its optimization goal is for GNNs trained on the condensed graph to perform similarly to those trained on the original. Graph condensation methods are typically categorized into two types: structured graph condensation, which generates both node features and graph structure, and structure-free methods, which only focus on synthesizing node features without explicitly constructing graph structures.

**Structured Graph Condensation.** These methods synthesize graph structures using a neural network that generates links between nodes based on their features. GCond (Jin et al., 2021) is the first such method, using a gradient matching loss between the original and condensed graphs, but its nested optimization loop limits efficiency. DosCond (Jin et al., 2022) introduced a more efficient one-step gradient match and a Bernoulli distribution for structure sampling. GCDM (Liu et al., 2022) generates smaller graphs with a distribution similar to the original graph, using a distribution matching loss measured by maximum mean discrepancy. SGDC reduces a graph set into a smaller set with fewer graphs via self-supervised representation matching. SGDD (Yang et al., 2024) incorporates the original graph structure through optimal transport. GDEM (Liu et al., 2023) aligns the eigenbasis of the condensed and original graphs to facilitate structure learning.

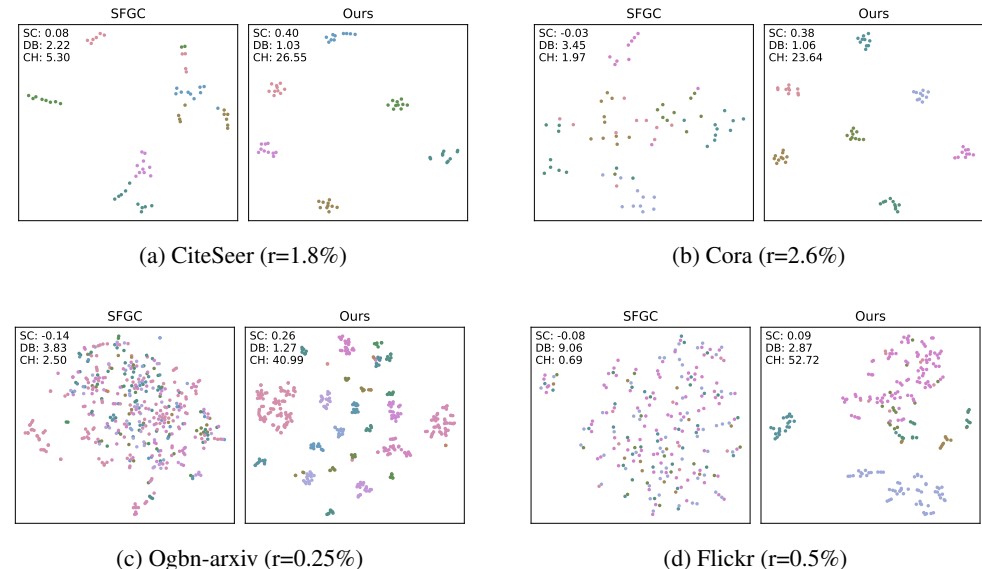

Figure 5: The t-SNE visualization of synthetic node features using GCN backbone. The node classes are represented by colors. The clustering metrics including Silhouette Coefficient (SC↑), Davies-Bouldin Index (DB↓), and Calinski-Harabasz Index (CH↑) are reported for each plot. The arrows ↑ and ↓ denote that a higher value indicates better clustering pattern for SC and CH, while a lower value indicates better clustering for DB.

**Structure-Free Graph Condensation.** These methods assume that structural information can be embedded directly into the synthetic node features, bypassing the need to generate graph structures. GCondX (Jin et al., 2021), a variant of GCond, focuses solely on feature learning via gradient matching without the inner loop. SFGC (Zheng et al., 2024) matches training trajectories with expert guidance, and GEOM (Zhang et al., 2024) adjusts the matching range for different node difficulties. To improve efficiency, GC-SNTK (Wang et al., 2024) replaces the inner loop using a kernel-based approach to synthesize a smaller graph efficiently.

**Graph Coarsening.** Graph coarsening methods (Cai et al., 2021; Loukas & Vandergheynst, 2018; Huang et al., 2021; Deng et al., 2019) reduce the graph's size by clustering nodes into super-nodes.

**Coreset Selection.** Coreset selection methods (Sener & Savarese, 2017; Welling, 2009; Wolf, 2011) condense the graph by selecting a subset of the original nodes and retaining the edges between them. K-Center (Sener & Savarese, 2017) trains a Graph Convolutional Network (GCN) (Kipf & Welling, 2016) on the original graph to generate embeddings, from which the k-nearest nodes are then sampled to form a subgraph.

Different from existing methods, our GCPA framework introduces a novel approach for graph condensation, simplifying the training process while enhancing performance. Our framework employs a streamlined, one-time precomputation and adaptation process that extracts and aligns features efficiently, avoiding the computationally expensive re-training phases seen in SOTA methods.

## 5 CONCLUSION

In this paper, we propose a new framework, GCPA, for graph condensation. It is efficient, consisting only of a one-time precomputation stage and a one-time adaptation learning stage. Compared to SOTA methods, our framework avoids costly repetitive re-training of models, achieving up to 1,890x faster training time than existing methods. Our framework is also effective, surpassing or matching the performance of the best baselines on 3 out of 7 benchmark datasets with just the one-time precomputation stage, and achieving SOTA results across all datasets with a further one-time adaptation learning stage. Our framework demonstrates that precomputation is a promising solution for efficient graph condensation, which is also flexible as it can be further enhanced through adaptation learning. In the future, we plan to explore more precomputation techniques for graph condensation.

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

# A    APPENDIX

## A.1    PERFORMANCE AND EFFICIENCY USING SGC BACKBONE

Figure 6 and 7 present node classification performance and efficiency comparison using SGC backbone, respectively.

Table 6: Node classification performance comparison using SGC backbone (mean±std). The best and second-best results are marked in bold and underlined, respectively. Ours (Pre.) is our precomputation-only variant. The Whole column represents the performance obtained by training on the whole dataset.

| Dataset | Ratio | K-Cen. | GCond | SGDD | GCDM | SFGC | GEOM | Ours | Whole |
|---|---|---|---|---|---|---|---|---|---|
| Citeseer | 0.9% | $52.7_{\pm0.0}$ | $71.9_{\pm0.6}$ | $71.1_{\pm0.1}$ | $66.0_{\pm2.2}$ | $65.2_{\pm0.3}$ | $60.1_{\pm0.2}$ | $\mathbf{72.3_{\pm0.5}}$ | |
| | 1.8% | $66.8_{\pm0.0}$ | $\underline{71.0_{\pm0.6}}$ | $69.9_{\pm0.1}$ | $66.7_{\pm0.0}$ | $67.0_{\pm0.8}$ | $65.2_{\pm0.2}$ | $\mathbf{72.7_{\pm0.3}}$ | $70.3_{\pm1.0}$ |
| | 3.6% | $68.1_{\pm0.0}$ | $\underline{72.5_{\pm1.2}}$ | $70.8_{\pm0.8}$ | $69.1_{\pm1.2}$ | $68.8_{\pm0.2}$ | $67.7_{\pm0.3}$ | $\mathbf{72.7_{\pm0.6}}$ | |
| Cora | 1.3% | $63.8_{\pm0.0}$ | $\underline{80.6_{\pm0.1}}$ | $62.4_{\pm5.5}$ | $77.0_{\pm0.4}$ | $73.8_{\pm1.5}$ | $69.2_{\pm1.2}$ | $\mathbf{80.9_{\pm0.7}}$ | |
| | 2.6% | $70.3_{\pm0.0}$ | $\underline{81.0_{\pm0.2}}$ | $80.8_{\pm0.4}$ | $78.9_{\pm1.0}$ | $77.5_{\pm0.1}$ | $69.6_{\pm1.5}$ | $\mathbf{81.5_{\pm0.6}}$ | $79.2_{\pm0.6}$ |
| | 5.2% | $77.1_{\pm0.0}$ | $80.9_{\pm0.4}$ | $\underline{81.4_{\pm0.4}}$ | $77.9_{\pm0.7}$ | $79.2_{\pm0.1}$ | $77.3_{\pm0.1}$ | $\mathbf{81.9_{\pm0.6}}$ | |
| PubMed | 0.08% | $70.5_{\pm0.1}$ | $75.9_{\pm0.7}$ | $\underline{76.4_{\pm0.9}}$ | $73.3_{\pm1.2}$ | $73.9_{\pm0.5}$ | $73.8_{\pm0.3}$ | $\mathbf{76.6_{\pm0.5}}$ | |
| | 0.15% | $75.8_{\pm0.0}$ | $75.2_{\pm0.0}$ | $\mathbf{78.0_{\pm0.3}}$ | $74.7_{\pm0.6}$ | $75.8_{\pm0.2}$ | $77.4_{\pm0.4}$ | $76.9_{\pm0.6}$ | $76.9_{\pm0.1}$ |
| | 0.3% | $75.7_{\pm0.0}$ | $75.7_{\pm0.0}$ | $76.1_{\pm0.1}$ | $\underline{76.5_{\pm1.1}}$ | $75.8_{\pm0.0}$ | $75.8_{\pm0.4}$ | $\mathbf{76.8_{\pm0.4}}$ | |
| Arxiv | 0.05% | $51.8_{\pm0.2}$ | $65.5_{\pm0.0}$ | $64.5_{\pm0.9}$ | $60.8_{\pm0.1}$ | $66.1_{\pm0.2}$ | $62.0_{\pm0.5}$ | $\mathbf{67.2_{\pm0.4}}$ | |
| | 0.25% | $58.2_{\pm0.0}$ | $66.5_{\pm0.5}$ | $66.4_{\pm0.3}$ | $62.7_{\pm0.9}$ | $\underline{66.7_{\pm0.3}}$ | $62.8_{\pm0.7}$ | $\mathbf{67.3_{\pm0.2}}$ | $68.8_{\pm0.0}$ |
| | 0.5% | $60.3_{\pm0.0}$ | $\underline{67.2_{\pm0.1}}$ | $66.9_{\pm0.3}$ | $62.4_{\pm0.2}$ | $66.4_{\pm0.3}$ | $63.6_{\pm0.3}$ | $\mathbf{67.3_{\pm0.1}}$ | |
| Products | 0.025% | $48.6_{\pm0.6}$ | $64.0_{\pm0.2}$ | $\underline{64.9_{\pm0.1}}$ | $57.7_{\pm0.2}$ | $62.2_{\pm0.1}$ | $61.1_{\pm0.4}$ | $\mathbf{65.0_{\pm0.5}}$ | |
| | 0.05% | $52.2_{\pm0.7}$ | $\underline{64.0_{\pm0.1}}$ | $62.3_{\pm0.2}$ | $58.2_{\pm0.3}$ | $62.2_{\pm0.2}$ | $62.4_{\pm0.2}$ | $\mathbf{65.1_{\pm0.4}}$ | $64.7_{\pm0.1}$ |
| | 0.1% | $55.4_{\pm0.4}$ | $\underline{64.4_{\pm0.4}}$ | $64.3_{\pm0.3}$ | $60.8_{\pm0.2}$ | $61.9_{\pm0.2}$ | $63.1_{\pm0.2}$ | $\mathbf{65.0_{\pm0.4}}$ | |
| Flickr | 0.1% | $34.5_{\pm0.1}$ | $43.7_{\pm0.5}$ | $43.6_{\pm0.3}$ | $40.3_{\pm0.0}$ | $\underline{45.3_{\pm0.7}}$ | $33.6_{\pm0.4}$ | $\mathbf{45.6_{\pm0.3}}$ | |
| | 0.5% | $36.1_{\pm0.0}$ | $42.2_{\pm0.2}$ | $41.6_{\pm1.6}$ | $40.8_{\pm0.1}$ | $\underline{45.7_{\pm0.4}}$ | $37.4_{\pm0.2}$ | $\mathbf{46.5_{\pm0.2}}$ | $44.2_{\pm0.0}$ |
| | 1% | $36.5_{\pm0.0}$ | $41.1_{\pm0.8}$ | $43.2_{\pm0.4}$ | $42.7_{\pm0.4}$ | $\underline{46.1_{\pm0.5}}$ | $38.1_{\pm0.2}$ | $\mathbf{46.8_{\pm0.2}}$ | |
| Reddit | 0.05% | $54.0_{\pm0.1}$ | $89.7_{\pm0.6}$ | $90.5_{\pm0.3}$ | $90.3_{\pm0.8}$ | $\underline{90.9_{\pm0.2}}$ | $59.4_{\pm1.5}$ | $\mathbf{91.5_{\pm0.7}}$ | |
| | 0.1% | $78.6_{\pm0.0}$ | $91.8_{\pm0.2}$ | $91.9_{\pm0.0}$ | $88.1_{\pm2.8}$ | $\mathbf{92.6_{\pm0.2}}$ | $81.7_{\pm0.7}$ | $\underline{92.6_{\pm0.1}}$ | $93.2_{\pm0.0}$ |
| | 0.2% | $83.8_{\pm0.0}$ | $92.1_{\pm0.3}$ | $86.3_{\pm5.6}$ | $91.7_{\pm0.2}$ | $\underline{92.6_{\pm0.3}}$ | $86.7_{\pm0.1}$ | $\mathbf{92.7_{\pm0.2}}$ | |

Table 7: Efficiency comparison using SGC backbone (total condensation time in seconds).

| Dataset | K-Center | GCond | SGDD | GCDM | SFGC | GEOM | Ours (Pre.) | Ours |
|---|---|---|---|---|---|---|---|---|
| Citeseer (r=1.8%) | 5 | 42 | 51 | 47 | 1,652 | 6,920 | 6 | 26 |
| Cora (r=2.6%) | 4 | 42 | 48 | 47 | 2,011 | 6,031 | 4 | 21 |
| PubMed (r=0.15%) | 4 | 34 | 204 | 42 | 7,555 | 22,201 | 5 | 40 |
| Arxiv (r=0.25%) | 6 | 283 | 1,485 | 242 | 78,586 | 84,356 | 20 | 71 |
| Products (r=0.05%) | 44 | 2,011 | 2,007 | 1,545 | 1,357,845 | 1,687,718 | 104 | 586 |
| Flickr (r=0.5%) | 5 | 177 | 300 | 258 | 99,254 | 19,202 | 23 | 56 |
| Reddit (r=0.1%) | 7 | 508 | 9,203 | 505 | 360,327 | 100,354 | 55 | 91 |

## A.2    IMPACT OF ADAPTATION LEARNING

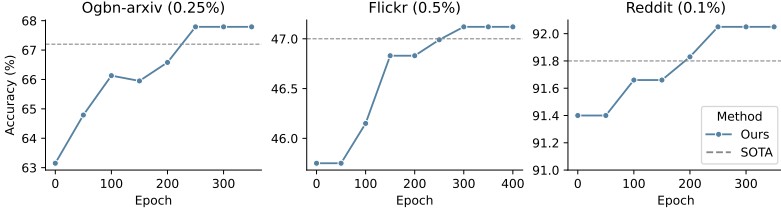

Figure 6: Impact of adaptation learning - performance after different number of adaptation learning epochs.

We demonstrate the impact of adaptation learning stage in Figure 6. On the presented large datasets, the precomputation stage (Epoch 0) produces condensed representations with sub-optimal performance. The adaptation learning further improves the representations by aligning them with the original node representations, achieving state-of-the-art performance after sufficient training epochs.

Table 8: Ablation study on the precomputation stage where we use randomly initialized features instead of precomputed features.

| Scheme | GCPA with Random Initialization | GCPA |
|---|---|---|
| Citeseer | 68.9 | **72.9** |
| Cora | 80.0 | **81.9** |
| PubMed | 73.5 | **80.5** |
| Arxiv | 66.0 | **67.7** |
| Products | 61.5 | **69.9** |
| Flickr | 43.8 | **47.1** |
| Reddit | 91.6 | **92.4** |

Table 9: Ablation study on the adaptation stage. We evaluate the precomputed features without adaptation and compare with the full framework.

| Dataset | SFGC | GEOM | GCPA w/o Adaptation | GCPA |
|---|---|---|---|---|
| Citeseer | 69.4 | 67.5 | 72.1 | **72.9** |
| Cora | 79.5 | 81.5 | 80.6 | **81.9** |
| PubMed | 78.1 | 79.7 | 79.7 | **80.5** |
| Arxiv | 67.2 | 65.8 | 64.6 | **67.7** |
| Products | 67.9 | 69.8 | 65.9 | **69.9** |
| Flickr | 47.0 | 45.2 | 45.4 | **47.1** |
| Reddit | 90.9 | 89.4 | 91.3 | **92.4** |

### A.3 ABLATION STUDY ON PRECOMPUTATION STAGE

We conduct an ablation study on the precomputation stage and present the results in Table 8. Specifically, we use randomly initialized synthetic features instead of sampled precomputed features. We detail the steps on aligning the labels of the condensed graph with the actual classes when features are initialized randomly. (1) **Initialization**: We map the randomly initialized features to the actual classes by assigning labels based on the original class distribution. The synthetic nodes are divided among the classes proportionally to their distribution on the original graph. This ensures that nodes associated with the same class are identified from the beginning. (2) **Adaptation**: In the adaptation stage, the synthetic nodes are optimized using a class-wise alignment loss to refine their features. This ensures that the synthetic features represent their respective classes more distinctly. The results illustrate the importance of the precomputation stage, which provides precomputed features that achieve better performance than randomly initialized features during the adaptation stage.

### A.4 ABLATION ON ADAPTATION STAGE

We conduct an ablation study concerning the feature adaptation module and present the results in Table 9. The presented results illustrate that while the precomputation stage yields competitive results on 4 out of 7 datasets (Citeseer, Cora, PubMed, and Reddit), the adaptation stage is crucial for further enhancing these precomputed representations, achieving superior results on the evaluated datasets.

### A.5 OBTAINING STRUCTURE-FREE FEATURES VIA PRECOMPUTATION

During the precomputation stage, we transform the raw features to structure-free features via precomputation. We use the derivations below to show that when using SGC as the backbone GNN, the precomputed features coupled with an identity adjacency matrix are equivalent to the raw features coupled with the original graph structures. We start by defining SGC network on the original graph:

$$\text{SGC}(\mathbf{X}, \mathbf{A}; \boldsymbol{\Theta}) = \left( \tilde{\mathbf{D}}^{-\frac{1}{2}} \tilde{\mathbf{A}} \tilde{\mathbf{D}}^{-\frac{1}{2}} \right)^K \mathbf{X} \boldsymbol{\Theta}, \qquad (6)$$

where $\mathbf{X}$ is the raw node features, $\tilde{\mathbf{A}} = \mathbf{A} + \mathbf{I}$ represents the adjacency matrix with self-loops, $\tilde{\mathbf{D}}$ denotes the degree matrix of $\tilde{\mathbf{A}}$, $K$ is the number of propagation layers, and $\boldsymbol{\Theta}$ is the weight matrix.

Then, we revisit the feature precomputation introduced in Equation 2 when $\alpha = 0$:

$$\mathbf{X}' = \left( \tilde{\mathbf{D}}^{-\frac{1}{2}} \tilde{\mathbf{A}} \tilde{\mathbf{D}}^{-\frac{1}{2}} \right)^K \mathbf{X}, \tag{7}$$

where $\mathbf{X}'$ denotes the precomputed features, which is the result of applying the same transformation as in the SGC but isolated from the learning weights $\boldsymbol{\Theta}$.

As a result, SGC with precomputed features and identity adjacency matrix becomes:

$$\text{SGC}(\mathbf{X}', \mathbf{I}; \boldsymbol{\Theta}) = \left( \tilde{\mathbf{D}}^{-\frac{1}{2}} \tilde{\mathbf{I}} \tilde{\mathbf{D}}^{-\frac{1}{2}} \right)^K \mathbf{X}' \boldsymbol{\Theta} = \mathbf{X}' \boldsymbol{\Theta} = \left( \tilde{\mathbf{D}}^{-\frac{1}{2}} \tilde{\mathbf{A}} \tilde{\mathbf{D}}^{-\frac{1}{2}} \right)^K \mathbf{X} \boldsymbol{\Theta}, \tag{8}$$

Therefore, we draw the equivalence between SGC computation on the original graph and the structure-free precomputed features:

$$\text{SGC}(\mathbf{X}', \mathbf{I}; \boldsymbol{\Theta}) = \text{SGC}(\mathbf{X}, \mathbf{A}; \boldsymbol{\Theta}) \tag{9}$$

The equivalence shows that although the original features and condensed features are differently distributed, they perform equivalently when coupled with their corresponding structures using the SGC backbone. Drawing inspiration from this equivalence under the SGC backbone, our framework focuses on initializing and refining features in the precomputed feature space, enabling effective training of message-passing GNNs on the structure-free condensed graphs.

