# OpenReview forum: "A Precompute-Then-Adapt Approach for Efficient Graph Condensation"
_ICLR.cc/2025/Conference — Submitted to ICLR 2025_

### Official Review · Reviewer_G3pa · 2024-10-27

**Soundness:** 2
**Presentation:** 2
**Contribution:** 2
**Rating:** 3
**Confidence:** 5

**Summary:**

This paper proposes a novel Precomputethen-Adapt graph condensation framework. The framework overcomes the efficiency limitation of the trajectory-matching condensation methods. By separating the condensation process into a one-time precomputation stage and a one-time adaptation learning stage, it achieves good experimental performance on the standard benchmark.

**Strengths:**

1. The approach this paper proposed reduces computational costs compared to previous gradient-matching or trajectory-matching methods.
2. The experimental results demonstrate the effectiveness of the method to some extent.

**Weaknesses:**

1. The novelty of this paper is limited. On the one hand, the so-called one-time precomputation stage in this paper, utilizing the mean value of sampled data features, is similar to some feature-matching dataset distillation methods; moreover, the story of "semantic" alignment was also proposed in [1]. On the other hand, the one-time adaptation learning process also just utilizes the commonly used graph contrastive loss.
2. The presentation of the paper is relatively weak. Basic concepts like Graph Condensation and Structure-Free Graph Condensation are repeatedly mentioned in both the introduction and related work sections, taking up a large portion of the main text. Additionally, the authors do not clearly explain the interrelationship between the two components.
3. In the experiments, the performance of the previous state-of-the-art methods is inconsistent with and significantly different from the performance reported in related technical papers. Was the baseline method carefully reproduced? Moreover, the method has an excessive number of hyperparameters and configurations, but the authors do not provide the hyperparameter settings for each dataset and condensation ratio or a principal tuning method, which undermines the effectiveness of the proposed method.
4. Since N' is the number of synthetic nodes, in eq 3, for i = 1, 2, . . . , N  ->  for i = 1, 2, . . . , N'.

[1] Diversified Semantic Distribution Matching for Dataset Distillation. ACM MM 2024

**Questions:**

Please refer to weaknesses.

---

> ### Author Response · Authors · 2024-11-21
>
> We sincerely appreciate your insightful comments.
>
> **Q1: The novelty of this paper is limited.**
>
> **A1:** While most graph condensation approaches rely on relatively complex and time-consuming gradient matching, our novelty lies in showing that (1) a different and more efficient initialization-only approach is sufficient to get SOTA performance with speedups of up to 2,455 times, and (2) contrastive loss can further enhance its performance, which has not been shown so far. Further, we believe that showing that simple and efficient methods are sufficient to achieve SOTA performance is beneficial for progress of the research field, as it shows that more complex / slower methods are unnecessary for achieving SOTA performance.
>
> Additionally, while our approach is simple, it is based on useful and non-obvious insights, of the importance of ensuring that our samples represent their class well, but also maintain diversity from one another. Our initialization method achieves this by randomly sampling a few nodes within a class to aggregate at a time, while our contrastive training achieves this by adjusting the samples to represent their class better. As it is not well understood what factors are needed for good performance in graph condensation, we believe this offers useful insights for the literature.
>
> **Q2: The presentation of the paper is relatively weak. Additionally, the authors do not clearly explain the interrelationship between the two components.**
>
> **A2:** **[Improve Paper Presentation]** We acknowledge the need for a clearer presentation and more precise explanations. Accordingly, we have reorganized the manuscript to reduce redundancy between the introduction and related work sections. Additionally, we have moved the related work section towards the end to emphasize our methodological innovations upfront.
>
> **[Interrelationship between Components]** Our graph condensation approach contains two components: the precomputation stage and the adaptation stage. The precomputation stage efficiently captures the structural and semantic features of the original graph. While effective, precomputation alone offers static initialization and there are opportunities for further performance improvements through adaptation learning. The Adaptation stage builds on the precomputed base, refining node features and enhancing model accuracy via representation adjustments, leading to improved alignment with original graph representations.
>
> **Q3: The performance of the previous state-of-the-art methods is inconsistent with and significantly different from the performance reported in related technical papers. Moreover, the method has an excessive number of hyperparameters and configurations, but the authors do not provide the hyperparameter settings for each dataset and condensation ratio or a principal tuning method.**
>
> **A3:** **[Reproduced Baseline Performance with Consistent Hyperparameters]** We acknowledge the differences between our reproduced baseline results and those reported in the original papers. To ensure fair comparisons, we uniformly set the hyperparameters for the GCN evaluators across all methods, following the practice adopted by the GCondenser benchmark [a]. This approach reduces variations due to differing GCN evaluators and allows for a more direct assessment of each method's effectiveness.
>
> Our hyperparameter settings maintain consistency with the original papers regarding the number of layers and the number of hidden units. Further, we standardized the dropout rate, weight decay, and learning rate for both our method and the reproduced baselines, to ensure all GCN evaluators are trained under the same conditions. Due to the word limit, please refer to the table below for a part of these settings:
>
> |Dataset|Hyperparameter|GCPA (Ours)|Reproduced Baselines (SFGC, GCond, etc.)|SFGC (Official)|GCond (Official)|
> |:-:|:-:|:-:|:-:|:-:|:-:|
> |Ogbn-arxiv|n_layers|2|2|2|2|
> ||n_hidden|256|256|256|256|
> ||dropout|0.5|0.5|0|0|
> ||weight_decay|0.0005|0.0005|0.005|0.0005|
> ||learning_rate|0.01|0.01|0.001|0.01|
> |Flickr|n_layers|2|2|2|2|
> ||n_hidden|256|256|256|256|
> ||dropout|0.5|0.5|0.6|0|
> ||weight_decay|0.0005|0.0005|0.07|0.0005|
> ||learning_rate|0.01|0.01|0.0005|0.01|
> |Reddit|n_layers|2|2|2|2|
> ||n_hidden|256|256|256|256|
> ||dropout|0.5|0.5|0|0|
> ||weight_decay|0.0005|0.0005|0.005|0.0005|
> ||learning_rate|0.01|0.01|0.01|0.01|
>
> **[Hyperparameter Setting and Tuning Method]** We provide specifics on our hyperparameter settings and tuning procedures in the Experimental Setup section. Specifically, for each dataset and condensation ratio, we conduct a grid search using the validation set, adjusting the hyperparameters within predefined ranges to optimize performance.
>
> **Q4: Since N' is the number of synthetic nodes, in eq 3, for i = 1, 2, . . . , N -> for i = 1, 2, . . . , N'.**
>
> **A4:** Thank you for pointing out the typo. We have fixed it in the revised manuscript.
>
> [a] GCondenser: Benchmarking Graph Condensation

---

> > ### Comment · Reviewer_G3pa · 2024-11-22
> >
> > Thanks for the response.
> >
> > I carefully review the feedback and the paper again. However, like Reviewer Luk2, I still cannot be convinced.
> >
> > The lack of novelty is still my concern. The approach is naive, and I still can not get the connections between the components. To me, they appear to be merely an initialization method and a commonly used contrastive learning approach to enhance representations.
> >
> > Moreover, the authors claim their method achieves SOTA; however, this is based on the fact that the performance of all previous methods is much lower than what has been reported in other technical papers. Since the authors applied grid search for the proposed method, how were the hyperparameters set for the previous methods during reproduction?

---

> > > ### Author Response · Authors · 2024-12-02
> > >
> > > Thank you for your detailed feedback and the thoughtful engagement with us.
> > >
> > > **Q5: The lack of novelty is still my concern. The approach is naive, and I still can not get the connections between the components. To me, they appear to be merely an initialization method and a commonly used contrastive learning approach to enhance representations.**
> > >
> > > **The novelty of our approach lies in the alignment mechanism applied to the precomputed feature space**, which enhances the learned representations beyond what standard initialization and existing contrastive techniques provide. As shown in Table 9, even without further training, sampling from the precomputed feature space achieves competitive performance, demonstrating that precomputation preserves key structural and semantic information. The subsequent adaptation stage aligns the condensed features with the precomputed original features, leading to performance that is comparable to training directly on the precomputed original features. This process is crucial for maintaining original graph information while improving downstream performance.
> > >
> > > **Q6: Moreover, the authors claim their method achieves SOTA; however, this is based on the fact that the performance of all previous methods is much lower than what has been reported in other technical papers. Since the authors applied grid search for the proposed method, how were the hyperparameters set for the previous methods during reproduction?**
> > >
> > > As described in **A3**, the difference in the baseline results is caused by uniformly setting the hyperparameters for the evaluator GNNs for fair comparisons.
> > >
> > > **We additionally present official results of GEOM (current SOTA) and our results when the hyperparameters of the evaluator GNNs are tunable** (dropout from $\{0,0.1,0.3,0.5,0.7,0.9\}$, weight decay from $\{0.005,0.001,0.0005,0.0001,0.00005,0.00001\}$, learning rate from $\{0.1,0.03,0.01,0.003,0.001,0.0003,0.0001\}$). We observe that our method achieves 8 SOTA results compared with 7 SOTA results achieved by GEOM, while achieving speedups of 96× to 2,455× (Table 3).
> > >
> > > |Dataset|Ratio|GEOM|Ours|
> > > |:---:|:---:|:---:|:---:|
> > > |Citeseer|0.9\%|73.0|**75.4**|
> > > ||1.8\%|74.3|**74.8**|
> > > ||3.6\%|73.3|**74.9**|
> > > |Cora|1.3\%|**82.5**|82.1|
> > > ||2.6\%|**83.6**|82.9|
> > > ||5.2\%|**82.8**|82.3|
> > > |Arxiv|0.05\%|65.5|**67.2**|
> > > ||0.25\%|**68.8**|67.7|
> > > ||0.5\%|**69.6**|68.1|
> > > |Flickr|0.1\%|**47.1**|47.0|
> > > ||0.5\%|47.0|**47.1**|
> > > ||1\%|**47.3**|47.1|
> > > |Reddit|0.05\%|91.1|**90.5**|
> > > ||0.1\%|91.4|**93.0**|
> > > ||0.2\%|91.5|**92.9**|
> > >
> > > **For the hyperparameters of the reproduced baselines, we strictly follow the baseline hyperparameter settings in Appendix A.3 of GCondenser benchmark [a] to present the reproduced results.** Concretely, the evaluator GNN hyperparameters (n_layers,n_hidden,dropout,weight_decay,learning_rate) are uniformly set as in **A3**. For the condensation models, the Adam optimizers are fixed with a weight decay of 0.5 and a learning rate of 0.01. The Bayesian hyperparameter sampler is applied to identify the optimal hyperparameters from the ranges below for each baseline method.
> > >
> > > | Hyperparameter | GCond, SGDD, GCDM |
> > > |:---:|:---:|
> > > | lr for adj | log_uniform(1e-6, 1.0) |
> > > | lr for feat | log_uniform(1e-6, 1.0) |
> > > | outer loop | 5 |
> > > | inner loop | 10 |
> > > | adj update steps | 10 |
> > > | feat update steps | 20 |
> > >
> > > | Hyperparameter | SFGC, GEOM |
> > > |:---:|:---:|
> > > | lr for feat | log_uniform(1e-6, 1.0) |
> > > | lr for student model | log_uniform(1e-3, 1.0) |
> > > | target epochs | start:0, end:800, step:10 |
> > > | warm-up epochs | start:0, end:100, step:10 |
> > > | student epochs | start:0, end:300, step:10 |
> > >
> > > By employing the hyperparameter searching scheme, we compare our method with baseline approaches on 5 common datasets (Citeseer, Cora, Ogbn-Arxiv, Flickr, Reddit) and 2 additional datasets (Pubmed and Ogbn-Products) to reduce variations due to differing GCN evaluators and allows for a more direct assessment of each method's effectiveness.

---

### Official Review · Reviewer_Luk2 · 2024-11-03

**Soundness:** 3
**Presentation:** 2
**Contribution:** 2
**Rating:** 5
**Confidence:** 5

**Summary:**

This paper propose a simple but effective and efficiency method for directly sample and merge node features (after message passing). Then using the alignment loss to enhance such features. The results are quite strong.

**Strengths:**

1. The performance of this work is comprehensive, typically including the PubMed and Products.
2. The effectiveness and efficiency are both impressive.

**Weaknesses:**

1. The methods used by the authors are not particularly novel, especially the use of contrastive loss for feature adaptation.
2. I find the structure of the condensed graph somewhat unclear; it appears that the condensed graph may lack structure. I suggest clarifying this aspect within the context.
3. The deeper rationale behind this straightforward method isn’t entirely apparent to me, yet it yields strong results. From my perspective, the pre-computation stage resembles the initialization of previous methods. I’m curious about the adaptation stage, which seems to primarily refine features selected in the pre-computation phase. Is it possible that the entire adaptation stage could be ablated?

**Questions:**

1. Please provide a deeper insight into your method, as I find this simple approach and its strong results intriguing. However, I remain unconvinced, as it seems to me like only a minor improvement on current initialization techniques, even prior to gradient-based or trajectory-based matching.
2. You might consider starting with some more ablation study, current one is quite short.

---

> ### Author Response · Authors · 2024-11-21
>
> We sincerely appreciate your insightful comments.
>
> **Q1: The methods used by the authors are not particularly novel, especially the use of contrastive loss for feature adaptation.**
>
> **A1:** **[Novelty]** While most graph condensation approaches rely on relatively complex and time-consuming gradient or trajectory computations, our novelty lies in showing that (1) a different and more efficient initialization-only approach is sufficient to get SOTA performance with speedups of up to 2,455 times, and (2) contrastive loss can further enhance its performance, which has not been shown so far. Further, we believe that showing that simple and efficient methods are sufficient to achieve SOTA performance is beneficial for progress of the research field, as it shows that more complex / slower methods are unnecessary for achieving SOTA performance.
>
> Additionally, while our approach is simple, it is based on useful and non-obvious insights, of the importance of ensuring that our samples represent their class well, but also maintain diversity from one another. Our initialization method achieves this by randomly sampling a few nodes within a class to aggregate at a time, while our adaptation learning achieves this by adjusting the samples to represent their class better. As it is not well understood what factors are needed for good performance in graph condensation, we believe this offers useful insights for the literature.
>
> **[Contrastive Loss as a Component]** The use of contrastive loss itself is not the novel aspect of our work. Instead, we emphasize a precompute-then-adapt framework, where the core innovation lies in first identifying good initial points based on the graph structure and then refining them. Contrastive loss is a common, versatile component we employ for the feature adaptation process.
>
> **Q2: I find the structure of the condensed graph somewhat unclear; it appears that the condensed graph may lack structure.**
>
> **A2:** Thank you for highlighting the ambiguity regarding the structure of the condensed graph. Indeed, the condensed graphs are structure-free, with nodes only connected to themselves, which aligns with many existing works focusing on node classification without requiring graph structures [a,b]. In the updated manuscript, we have clarified this point in the introduction.
>
> **Q3: Is it possible that the entire adaptation stage could be ablated?**
>
> **A3:** We appreciate your interest in the adaptation stage of our method. As shown in the table below, the pre-computation stage alone achieves performance close to the best existing methods. However, the adaptation stage is crucial, as it further refines the features and enables our method to surpass most SOTA results across the datasets. We have included these results in the updated paper, along with a new table (Table 9).
>
> |Dataset|SFGC|GEOM|GCPA w/o Adaptation|GCPA|
> |:---:|:---:|:---:|:---:|:---:|
> |Citeseer|69.4|67.5|72.1|**72.9**|
> |Cora|79.5|81.5|80.6|**81.9**|
> |PubMed|78.1|79.7|79.7|**80.5**|
> |Arxiv|67.2|65.8|64.6|**67.7**|
> |Products|67.9|69.8|65.9|**69.9**|
> |Flickr|47.0|45.2|45.4|**47.1**|
> |Reddit|90.9|89.4|91.3|**92.4**|
>
> **Q4: I remain unconvinced, as it seems to me like only a minor improvement on current initialization techniques, even prior to gradient-based or trajectory-based matching.**
>
> **A4:** While our method is simple, it is fundamentally different from gradient matching or trajectory matching approaches. Instead of relying on repetitive and computationally intensive gradient or trajectory calculations, we employ a two-stage process of precomputation and adaptation. This allows us to efficiently condense graph data by ensuring that our samples both accurately represent their classes and maintain diversity among themselves. This strategy enables us to achieve comparable or superior performance to state-of-the-art methods, but with significantly reduced computational overhead.
>
> **Q5: You might consider starting with some more ablation study, current one is quite short.**
>
> **A5:** Apart from the added ablation study introduced in **A3**, we present an additional ablation study on the precomputation stage. We use randomly initialized synthetic features instead of sampled original features. The results shown below illustrates the importance of our precomputation stage, where the precomputed features achieve better performance during the adaptation stage than the randomly initialized features. We have included these results in Table 8 of the updated paper.
>
> |Scheme|GCPA with Random Initialization|GCPA|
> |:---:|:---:|:---:|
> |Citeseer|68.9|72.9|
> |Cora|80.0|81.9|
> |PubMed|73.5|80.5|
> |Arxiv|66.0|67.7|
> |Products|61.5|69.9|
> |Flickr|43.8|47.1|
> |Reddit|91.6|92.4|
>
> [a] Navigating Complexity: Toward Lossless Graph Condensation via Expanding Window Matching
>
> [b] Structure-free Graph Condensation: From Large-scale Graphs to Condensed Graph-free Data

---

> > ### Comment · Reviewer_Luk2 · 2024-11-21
> > **Discussion about revised paper**
> >
> > Dear authors,
> >
> > Thanks for your detailed rebuttal, could I ask if the revised paper you have updated or you will update after rebuttal? That's because you mentioned a lot you'll update, but I don't know if I need to print your paper again for reference?
> >
> > Best,
> > Reviewer Luk2

---

> > > ### Author Response · Authors · 2024-11-21
> > >
> > > Dear Reviewer Luk2,
> > >
> > > We would like to inform that we have uploaded an updated version of our paper's PDF. All modifications and additions are highlighted in blue for easier identification and review.
> > >
> > > Thank you for your attention and for your valuable feedback which has contributed to these revisions.
> > >
> > > Best regards,
> > > Authors of Submission 5832

---

> > > > ### Comment · Reviewer_Luk2 · 2024-11-22
> > > >
> > > > **1. Why does the precomputation component work?**
> > > >
> > > > The precomputation component, as I understand it, involves two main steps: (1) performing message passing, akin to SGC (Simplified Graph Convolution), and (2) randomly sampling nodes within each class and calculating the mean representation of their features. Please correct me if my understanding is off.
> > > >
> > > > I must say, I am impressed by the performance of this component in your ablation studies (thank you for including that), but I do have a concern. It seems to me that pre-aggregating features in this manner might create a significant out-of-distribution (OOD) issue for the downstream GNNs. This is because the precomputed features are inherently different from the raw test features. Moreover, with this approach, there’s a risk that the final representation may become over-smoothed, which could diminish its effectiveness.
> > > >
> > > > This concern stems from what might be a misunderstanding of your work. Could you provide a deeper explanation of why the precomputation step avoids these pitfalls, beyond just presenting empirical results? Simpler methods like this often require more detailed theoretical analysis to establish their robustness.
> > > >
> > > > **2. The last additional table appears to be misunderstood**
> > > >
> > > > If the features are initialized randomly, how do you align the labels of the condensed graph with the actual classes? Specifically, random initialization creates arbitrary features, and the goal of adaptation is to cluster these features into coherent groups. For example, in the Cora dataset, you would aim to form 7 distinct groups. However, how can you guarantee that, say, the first group of features corresponds to the first class label in the original dataset?
> > > >
> > > > This seems like a critical gap that could lead to my confusion about how the condensed features relate to the original dataset. Could you clarify this mapping process?
> > > >
> > > > **About novelty and impact:** I now recognize the contribution of this work in pointing out the cumbersome and less-effective aspects of previous methods. However, I can really cannot be convinced by current method, in contract, I think it might perform worse than previous methods. **It is actually an interesting problem, I encourage author to check the deep reason in this discussion peroid or later. It would greatly impact to this research topic.**
> > > >
> > > > I'll raise soundness to 3, for other scores, I think they can fairly represent the work by now.

---

> > > > > ### Author Response · Authors · 2024-11-25
> > > > >
> > > > > Thank you for your detailed feedback and the thoughtful engagement with us.
> > > > >
> > > > > **Q6: Why does the precomputation component work? Pre-aggregating features in this manner might create a significant out-of-distribution (OOD) issue for the downstream GNNs. There’s a risk that the final representation may become over-smoothed.**
> > > > >
> > > > > **Precomputation alleviates OOD issues,** since in structure-free settings [a,b], a condensed graph has no edges and cannot leverage neighbor information as on original graphs. To address this limitation, precomputation incorporates neighbor information directly into the condensed nodes. For instance, consider the SGC model:
> > > > >
> > > > > On original: $SGC(X,A;\Theta)=MLP(A^kX;\Theta).$
> > > > >
> > > > > On condensed: $SGC(X',I;\Theta)=MLP(X';\Theta)=MLP(\text{precompute}(X,A);\Theta)=MLP(\text{sample}(A^kX);\Theta),$
> > > > > where the sampler takes a subset of original nodes for condensation.
> > > > >
> > > > > The equations illustrate that precomputation allows the SGC applied on the condensed graph, $SGC(X',I;\Theta)$, to function comparably to the SGC on the original graph, $SGC(X,A;\Theta)$. This is because the inputs to the MLP are from the same distribution, either $A^kX$ or $\text{sample}(A^kX)$, thereby alleviating OOD concerns.
> > > > >
> > > > > While this approach is straightforward for SGC, more complex GNNs may not approximate neighbor information as effectively. Nonetheless, precomputation still enhances performance by enabling GNNs on condensed graphs to access neighbor information from corresponding nodes on original graphs.
> > > > >
> > > > > **Alleviating over-smoothing.** The potential risk of over-smoothing is indeed an important aspect to consider in GNNs. To mitigate the risk, we employ few aggregation layers (1 to 4 layers) to maintain feature distinction between nodes. We present an extended parameter analysis below. To achieve good performance, we only need few aggregation layers (e.g., 2 layers), which suffer less from over-smoothing.
> > > > >
> > > > > |Aggregation Layers|Ogbn-Arxiv|Flickr|Reddit|
> > > > > |:---:|:---:|:---:|:---:|
> > > > > |1|62.0|47.1|**92.5**|
> > > > > |2|**67.7**|47.1|92.0|
> > > > > |3|60.9|**47.2**|92.2|
> > > > > |4|61.8|47.1|92.4|
> > > > >
> > > > > **Q7: The last additional table appears to be misunderstood. If the features are initialized randomly, how do you align the labels of the condensed graph with the actual classes?**
> > > > >
> > > > > In the ablation study presented in Table 8, GCPA with Random Initialization is a variant of our GCPA approach. This variant assesses the impact of precomputed features by replacing them with random features, while retaining their labels that are initially assigned by GCPA.
> > > > >
> > > > > During the adaptation phase, these random features are encoded by the adaptation module to generate class-representative node representations. The adaptation module selectively samples node features from the original graph, aligning the adapted representations more closely with those of their respective classes. We have updated the paper to further detail this process in Appendix A.3.
> > > > >
> > > > > **Q8: About novelty and impact: I think it might perform worse than previous methods. I encourage author to check the deep reason.**
> > > > >
> > > > > **[Novelty and Insights]** We introduce a novel graph condensation framework that streamlines structure and semantic-based feature precomputation with adaptation learning. We highlight the non-obvious insight that by jointly deploying the two components, we can efficiently and effectively condense large graphs to compact, structure-free representations. The performance and efficiency results validate that our approach offers a unique advantage by reducing computational demands without compromising accuracy.
> > > > >
> > > > > **[Underlying Reason for Effectiveness]** The effectiveness of pre-computation methods in node classification tasks may stem from their ability to incorporate extensive semantic and structural information from original graphs into condensed ones. Evidence of this is shown by our method's ability to match or exceed state-of-the-art performance on 3 out of 7 datasets, even without an adaptation stage. Additionally, aligning condensed features with the original graph’s precomputed features enhances their representation of class characteristics. This is supported by t-SNE visualizations (Figure 5), which demonstrate distinct class boundaries within the condensed features. These features enable GNNs to effectively extract knowledge from condensed graphs, improving performance on original graphs.
> > > > >
> > > > > **[Outstanding Performance]** The empirical results (Tables 2 and 6) highlight our method's leading performance across benchmarks, achieving state-of-the-art results in 40 out of 42 settings when tested with GCN and SGC backbones. Furthermore, our method is efficient, offering speedups of 96× to 2,455× compared to trajectory-based approaches, addressing scalability issues prevalent in prior methods. This efficiency and performance underscore the potential impact of our precompute-then-adapt approach on advancing future graph condensation researches.

---

> > > > > > ### Comment · Reviewer_Luk2 · 2024-11-25
> > > > > >
> > > > > > Thanks for authors’ detailed response!
> > > > > >
> > > > > > > OOD problem on precomutation stage
> > > > > > >
> > > > > >
> > > > > > Okay, I now have a better understanding of this point. It simulates the SGC by allowing the downstream model to directly learn from the aggregated features without edges, which enables generalization to the test phase where edges are present.
> > > > > >
> > > > > > However, **I don’t think it’s intuitively clear whether this strategy truly alleviates the OOD problem.** For instance, if the downstream model is GAT or a graph transformer, which needs to learn how to aggregate information, your assumption might not hold in this case. Frankly, more evidence is needed to demonstrate that the **precomputed aggregation combined with a condensed, edge-less graph** performs better than **simply using the naive features with a condensed graph which has edges.**
> > > > > >
> > > > > > I think it would be beneficial to explore this point further in your work to provide a more coherent and convincing argument.
> > > > > >
> > > > > > > Table 8
> > > > > > >
> > > > > >
> > > > > > Thank you for the clarification. I now understand that you align the labels by still using the assigned nodes. However, an interesting question remains regarding the OOD issue. If you’re enhancing the class-wise differences using contrastive learning, how can you be certain that the learned features are IID (independently and identically distributed) with respect to the original dataset?
> > > > > >
> > > > > > In other words, when you initialize 7 groups of random features, as I now understand it, you use contrastive loss to regenerate these groups, increasing the inter-group difference. **My question is, do the features in the 1st generated group share the same distribution as the real 1st class features?**
> > > > > >
> > > > > > I checked Appendix A.3, but it didn’t help on address my concern. I am still unsure how the randomly generated features can align with the real distribution. Did I use the wrong version?
> > > > > >
> > > > > > ```latex
> > > > > > We conduct an ablation study on the precomputation stage in Table 8. Specifically, we use randomly
> > > > > > initialized synthetic features instead of sampled precomputed features. The results illustrate the
> > > > > > importance of the precomputation stage, which provides precomputed features that achieve better
> > > > > > performance than randomly initialized features during the adaptation stage.
> > > > > > ```
> > > > > >
> > > > > > > Novelty and impact
> > > > > > >
> > > > > >
> > > > > > I would like to claim that **I appreciate the results you have reported**. There is no need to repeat this again, which is why I believe it is crucial to explain the **underlying reasons behind these results**. The current precomputation and adaptation methods appear to be **more of a technical and incremental contribution**. Most importantly, I still cannot fully understand why they lead to such significant improvements.
> > > > > >
> > > > > > I appreciate the authors’ efforts and believe this paper has the potential for significant impact. However, you must first convince the reader **how your two relatively simple operations address the issues present in current methods**. Merely reporting the results is insufficient. Additionally, the vague description of the “compact, structure-free” representation is not entirely convincing.
> > > > > >
> > > > > > It is welcome for you to address my concerns during this period, but I believe that with a more detailed and motivated reorganization, it will be a great work can be accepted by any conference.
> > > > > >
> > > > > > Best,
> > > > > >
> > > > > > Reviewer Luk2

---

> > > > > > > ### Author Response · Authors · 2024-12-02
> > > > > > >
> > > > > > > We appreciate your detailed feedback and the thoughtful engagement with us.
> > > > > > >
> > > > > > > **Q9: I don’t think it’s intuitively clear whether this strategy truly alleviates the OOD problem, for instance, if the downstream model is GAT or a graph transformer.
> > > > > > > More evidence is needed to demonstrate that the precomputed aggregation combined with a condensed, edge-less graph performs better.**
> > > > > > >
> > > > > > > **We acknowledge that the OOD issue is not fully resolved for all architectures, particularly for graph transformers.** As discussed in **Q6**, our approach alleviates the OOD problem for message-passing GNNs, such as SGC. However, addressing the challenge of cross-architecture transferability remains an open issue. While GAT is comparable to other message-passing GNNs (Figure 10 of GC4NC [c] and our Table 5), existing graph condensation techniques struggle to achieve similar results with graph transformers (Figure 10 of GC4NC [c]).
> > > > > > >
> > > > > > > Nevertheless, as presented in Table 5, our method outperforms baseline approaches on benchmark datasets even in the presence of OOD. A potential reason is that in node classification tasks, the $A^kX$ features are sufficiently informative, so that our approach remains relatively robust under OOD conditions.
> > > > > > >
> > > > > > > **Theorectical evidence for removing structures.** Regarding your concern about the effectiveness of structure-free graphs compared to structured graphs, we follow SFGC [b] to argue that, from a learning perspective, removing graph structures can be a valid approach for graph condensation. Specifically, the goal of a GNN trained on graph $\{X',A',Y'\}$ is to learn the conditional probability for node classification:
> > > > > > > $$\begin{aligned}
> > > > > > > Q(Y' \mid X', A') &= \sum_{A' \in \psi(X')} Q(Y' \mid X', A') Q(A' \mid X') \\
> > > > > > > &= \sum_{A' \in \psi(X')} \frac{Q(X', A', Y')}{Q(X')} \\
> > > > > > > &= Q(Y' \mid X'),
> > > > > > > \end{aligned}$$
> > > > > > > where $\psi(X')$ is a structure-learning module [d]. When the structures are learned from features ($A' \in \psi(X')$), the goal is to solve the posterior $Q(Y' \mid X')$. In this sense, removing graph structures for condensation is reasonable.
> > > > > > >
> > > > > > > We also address the vague "compact, structure-free representation". We clarify that these structure-free features are compact because they integrate implicit topological information.
> > > > > > >
> > > > > > > **Empirical evidence for removing structures.** In Table 6, we show that structured condensation approaches (K-Center, GCond, SGDD, and GCDM) perform consistently worse than structure-free approaches. This aligns with the theoretical analysis that structure learning can be removed.
> > > > > > >
> > > > > > > **Q10: If you’re enhancing the class-wise differences using contrastive learning, how can you be certain that the learned features are IID with respect to the original dataset?**
> > > > > > >
> > > > > > > **We clarify that the learned features are IID with respect to the precomputed original features**, but not the raw original features, as also explained in our response to **Q6**. To formalize this, we consider the information preservation during the feature condensation process.
> > > > > > >
> > > > > > > Let $p(\mathbf{H} | y = c)$ represent the precomputed original feature distribution conditioned on class $c$, and $q(\mathbf{Z'} | y = c)$ the condensed features. The objective is to minimize the divergence between them, ensuring that the condensed features match the statistical properties of the precomputed original features. By employing the loss function in Equation 5, we implicitly minimize the Kullback-Leibler (KL) divergence between $𝑝$ and $𝑞$ for each class [e]:
> > > > > > >
> > > > > > > $$\mathrm{KL}\big( p(\mathbf{H} | y = c) \parallel q(\mathbf{Z'} | y = c) \big) = \int p(\mathbf{H} | y = c) \log \frac{p(\mathbf{H} | y = c)}{q(\mathbf{Z'} | y = c)} d\mathbf{H}.$$
> > > > > > >
> > > > > > > Minimizing this KL divergence ensures that the condensed features $\mathbf{Z'}$ are statistically similar to the original features $\mathbf{H}$ within each class. As a result, the model learns to generate **independent** features that are **identically distributed** with respect to the original class-specific precomputed features, which is crucial for maintaining class-specific information and effective classification.
> > > > > > >
> > > > > > > **Q11: I believe it is crucial to explain the underlying reasons behind these results.**
> > > > > > >
> > > > > > > **The significant improvements in performance can be attributed to the effectiveness of alignment in the precomputed feature space, a novel contribution in the literature.** As mentioned in response to **Q10**, the representations are aligned such that the condensed features closely correspond to the original precomputed features. This alignment ensures that the learned features maintain much information from the precomputed feature space, leading to performance that is comparable to training directly on the original precomputed features.
> > > > > > >
> > > > > > > [c] GC4NC: A Benchmark Framework for Graph Condensation on Node Classification with New Insights
> > > > > > >
> > > > > > > [d] Graph Condensation for Graph Neural Networks
> > > > > > >
> > > > > > > [e] Contrastive Representation Distillation

---

> > > > > > > > ### Comment · Reviewer_Luk2 · 2024-12-02
> > > > > > > >
> > > > > > > > Thank you for the authors' detailed rebuttal, which has addressed several of my concerns:
> > > > > > > >
> > > > > > > > > About the **Precomputation Stage**
> > > > > > > >
> > > > > > > > Your theoretical evidence, which demonstrates that your method approximates existing non-precompute methods, does not address the core issue. The key question remains: why not avoid the OOD problem entirely by using existing methods, rather than introducing your method and subsequently mitigating the OOD issue? I suggest It would be more compelling to provide evidence that your method is **better** than existing non-precompute methods.
> > > > > > > >
> > > > > > > > While your empirical evidence is strong, it still lacks a deeper understanding. You metioned methods like K-Center, GCond, SGDD, and GCDM, involving at least three distinct designs. To strengthen your argument, **I suggest starting with their no-structure variants to build a more coherent findings**. For instance, the GCond-X and GCDM-X variants discussed in their respective papers offer some insights, and you could leverage these to establish stronger connections and uncover new insights. The current direct comparisons leave too many uncertain variables.
> > > > > > > >
> > > > > > > > My concerns about this aspect remain. Thank you for the clarification provided thus far.
> > > > > > > >
> > > > > > > > > About the **Adaptation Stage**
> > > > > > > >
> > > > > > > > To confirm, is the **target feature distribution also precomputed**? I could not find a clear statement to this effect in Lines 242–255 of the manuscript. If this is the case, I believe it **constitutes a critical ambiguity** in your writing. I strongly recommend revising this section to make it explicit.
> > > > > > > >
> > > > > > > > My concerns about this part have been resolved. Thank you for the clarification.
> > > > > > > >
> > > > > > > > ---
> > > > > > > > I will maintain my score and discuss further with other reviewers and chairs during the later review period.

---

> ### Comment · Reviewer_Luk2 · 2024-12-02
>
> Thank you for the authors' detailed rebuttal, which has addressed several of my concerns:
>
> > About the **Precomputation Stage**
>
> Your theoretical evidence, which demonstrates that your method approximates existing non-precompute methods, does not address the core issue. The key question remains: why not avoid the OOD problem entirely by using existing methods, rather than introducing your method and subsequently mitigating the OOD issue? I suggest It would be more compelling to provide evidence that your method is **better** than existing non-precompute methods.
>
> While your empirical evidence is strong, it still lacks a deeper understanding. You metioned methods like K-Center, GCond, SGDD, and GCDM, involving at least three distinct designs. To strengthen your argument, **I suggest starting with their no-structure variants to build a more coherent findings**. For instance, the GCond-X and GCDM-X variants discussed in their respective papers, and you could leverage these to establish stronger connections and uncover new insights. The current direct comparisons leave too many uncertain variables. (Just suggestion for your later revision, not the new experiments in this period)
>
> My concerns about this aspect remain. Thank you for the clarification provided thus far.
>
> > About the **Adaptation Stage**
>
> To confirm, is the **target feature distribution also precomputed**? I could not find a clear statement to this effect in Lines 242–255 of the manuscript. If this is the case, I believe it **constitutes a critical ambiguity** in your writing. I strongly recommend revising this section to make it explicit.
>
> My concerns about this part have been resolved. Thank you for the clarification.
>
> ---
> I will maintain my score and discuss further with other reviewers and chairs during the later review period.

---

> > ### Author Response · Authors · 2024-12-03
> >
> > We appreciate your detailed feedback and thoughtful engagement with our work. We would like to address your concerns raised in the latest review.
> >
> > **Q12: The key question remains: why not avoid the OOD problem entirely by using existing methods, rather than introducing your method and subsequently mitigating the OOD issue? I suggest It would be more compelling to provide evidence that your method is better than existing non-precompute methods.**
> >
> > We apologize if our previous response caused any confusion. To clarify, in **Q9**, we stated that the OOD problem remains a challenge for all existing methods, including ours. The OOD issue is particularly evident when using graph transformers as the downstream model. All existing graph condensation techniques face degraded performance with graph transformers, as demonstrated in Figure 10 of the GC4NC benchmark [c].
> >
> > The root of this issue lies in the fact that most baseline approaches rely on specific GNN models (e.g., GCN) during graph condensation, which inherently introduces OOD challenges when transferring the condensed graphs to different GNN models. Specifically:
> > - K-Center uses embeddings from a particular GNN model to select a core set of features
> > - GCond and SGDD rely on specific GNN models to compute and match gradients
> > - GCDM depends on embeddings and distribution matching from a particular GNN model
> > - SFGC and GEOM use specific GNN models to track and match the trajectory of GNN parameters
> >
> > Given this reliance on specific GNN architectures, the transfer of condensed graphs across different models inevitably leads to OOD problems.
> >
> > Despite these challenges, our empirical results, as shown in Table 5, demonstrate the superiority of our method over baseline approaches. We hypothesize that the key advantage of our approach is the robustness of the learned features in the precomputed feature space. This allows for better transferability across different GNN architectures, thereby mitigating the OOD issue relatively well.
> >
> > **Q13: While your empirical evidence is strong, it still lacks a deeper understanding. You metioned methods like K-Center, GCond, SGDD, and GCDM, involving at least three distinct designs. To strengthen your argument, I suggest starting with their no-structure variants to build a more coherent findings. For instance, the GCond-X and GCDM-X variants discussed in their respective papers, and you could leverage these to establish stronger connections and uncover new insights. The current direct comparisons leave too many uncertain variables. (Just suggestion for your later revision, not the new experiments in this period)**
> >
> > Thank you for the insightful comment. The comparison between structure-free and structured condensation is indeed a intriguing area of inquiry. As discussed in Q9, our current hypothesis is that removing condensed structures reduces the number of learnable parameters while preserving the core learning objectives. This, in turn, may lead to easier optimization and better performance. We agree that exploring this idea more thoroughly in future research could provide valuable insights, and we plan to investigate this further in subsequent work.
> >
> > **Q14: To confirm, is the target feature distribution also precomputed? I could not find a clear statement to this effect in Lines 242–255 of the manuscript. If this is the case, I believe it constitutes a critical ambiguity in your writing. I strongly recommend revising this section to make it explicit.**
> >
> > Thank you for your observation. Yes, the target feature distribution is indeed precomputed. This is clarified in Equation 5, where we adapt the condensed features using the precomputed features $H$, as previously defined in Equation 2. We acknowledge that this point could be more explicit in the text and appreciate your feedback. In the revised manuscript, we will address this ambiguity by consistently using the term "precomputed features" throughout the relevant sections.

---

### Official Review · Reviewer_9kFh · 2024-11-04

**Soundness:** 3
**Presentation:** 3
**Contribution:** 3
**Rating:** 8
**Confidence:** 5

**Summary:**

The paper introduces "Precompute-then-Adapt," a fast and efficient framework for graph condensation in large-scale Graph Neural Networks (GNNs). Unlike traditional methods that require repeated, costly retraining, this approach divides the process into a one-time precomputation phase and a one-time adaptation stage. This novel framework achieves competitive or superior results on several benchmarks, reducing condensation time and making GNNs far more practical for large-scale applications.

**Strengths:**

S1: Efficiency: The proposed"Precompute-then-Adapt" framework does significantly reduce the graph condensation process’s time,  and it effectively solves the repeat training consumption on the large-scale graph for existing structure-free methods, which sounds rational to me.

S2: High Performance with Simplified Methodology: The framework achieves expressive performance on several benchmark datasets with various condensation ratios, even with just the initial precomputation phase. Its one-time precompute strategy with both structure aggregation and semantic aggregation looks simple but efficient, making the proposed methodology practical.

S3: Clarity: The paper is well-structured, with a clear and logical flow that makes it easy to read and follow. All the experiential results are clearly demonstrated.

**Weaknesses:**

I do have some concerned questions in terms of methodology and experiments:

W1: Unlike methods such as GCOND-X and SFGC, which mimic the GNN's learning behavior, the proposed method appears independent of specific GNN backbones, focusing instead on operations at the graph data level without involving models (one-time precomputation is not related to trained GNNs). This raises a question: how does the approach ensure similar performance on the original test graph set when no explicit GNN model-based constraints are applied?

W2: For the ablation study, it only shows w/o structure and w/o semantics, what about only with precompute features but without the feature adaption module and contrastive learning?

W3: Also for the ablation study, what is w/o both mean? does that mean using random initialized features to input the feature adaption module?

**Questions:**

See weakness above.

---

> ### Author Response · Authors · 2024-11-21
>
> We sincerely appreciate your insightful comments.
>
> **Q1: Unlike methods such as GCOND-X and SFGC, which mimic the GNN's learning behavior, the proposed method appears independent of specific GNN backbones, focusing instead on operations at the graph data level without involving models (one-time precomputation is not related to trained GNNs). This raises a question: how does the approach ensure similar performance on the original test graph set when no explicit GNN model-based constraints are applied?**
>
> **A1:** The proposed method ensures strong performance on the original test graph set by enhancing the graph data itself, which benefits any GNN architecture. By focusing on data-level operations, i.e., refining node features, it universally improves the quality of the input graph without relying on specific model constraints. These enhancements make the dataset more suitable for learning across different GNNs, ensuring similar or even improved performance. This model-agnostic effectiveness is demonstrated in Table 5, showing strong cross-architecture transferability results.
>
> **Q2: For the ablation study, it only shows w/o structure and w/o semantics, what about only with precompute features but without the feature adaption module and contrastive learning?**
>
> **A2:** The ablation study has been expanded to include an evaluation of the model using only precomputed features without the feature adaptation module and contrastive learning. The results indicate that while the model with only precomputed features performs adequately on some datasets, it consistently underperforms compared to the full model that includes the feature adaptation module and contrastive learning. Notably, we observed a decrease in accuracy by 10.0% on Ogbn-Arxiv and 7.5% on Ogbn-Products when omitting these components. This highlights the importance of the feature adaptation module in enhancing model performance.
>
> We have included these results in the appendix of the updated paper, along with a new table (Table 9), providing a clearer comparison of the different setups.
>
> |Dataset|Ratio|GCPA w/o Adaptation|GCPA|
> |:---:|:---:|:---:|:---:|
> |Citeseer|0.9\%|72.1|**72.4**|
> ||1.8\%|72.1|**72.9**|
> ||3.6\%|72.7|**72.8**|
> |Cora|1.3\%|80.3|**81.4**|
> ||2.6\%|80.6|**81.9**|
> ||5.2\%|80.8|**82.4**|
> |PubMed|0.08\%|79.5|**80.2**|
> ||0.15\%|79.7|**80.5**|
> ||0.3\%|79.3|**81.6**|
> |Arxiv|0.05\%|60.5|**67.2**|
> ||0.25\%|64.6|**67.7**|
> ||0.5\%|65.5|**68.1**|
> |Products|0.025\%|64.1|**69.3**|
> ||0.05\%|65.9|**69.9**|
> ||0.1\%|67.7|**71.3**|
> |Flickr|0.1\%|44.4|**47.0**|
> ||0.5\%|45.4|**47.1**|
> ||1\%|45.4|**47.1**|
> |Reddit|0.05\%|**90.5**|**90.5**|
> ||0.1\%|91.3|**92.4**|
> ||0.2\%|91.4|**92.9**|
>
> **Q3: Also for the ablation study, what is w/o both mean? does that mean using random initialized features to input the feature adaption module?**
>
> **A3:** In the ablation study, "w/o both" means without both structure-based and semantic-based precomputations. Instead, we input randomly selected original features into the adaptation module.

---

> ### Comment · Reviewer_9kFh · 2024-11-22
>
> Thanks for your response. Given more ablation study you provided, I would like to raise my score.

---

> > ### Author Response · Authors · 2024-11-22
> >
> > Dear Reviewer 9kFh,
> >
> > We sincerely appreciate your constructive feedback, and we are delighted that our responses have effectively addressed your concerns. Your recognition of our efforts in the paper is greatly appreciated!
> >
> > Warm regards,
> > The Authors of Submission 5832

---

### Meta-Review · Area_Chair_etqu · 2024-12-11

**Metareview:**

This paper introduces a Precompute-then-Adapt graph condensation framework aimed at addressing the high computational costs of previous methods. However, based on the feedback from Reviewer G3pa and Reviewer Luk2, there are significant concerns regarding the limited novelty of the proposed framework, the lack of rigor in comparisons with other methods, and the subpar performance on large datasets compared to state-of-the-art approaches. Given these issues, the submission does not meet the standards for acceptance at ICLR.

**Additional Comments On Reviewer Discussion:**

The paper lacks novelty, provides insufficiently rigorous comparisons with other methods, and demonstrates subpar performance on large datasets.

---

### Decision · Program_Chairs · 2025-01-22

Reject